# INCENTIVES IN FEDERATED LEARNING WITH HETEROGENEOUS AGENTS

**Ariel D. Procaccia**
Harvard University

**Han Shao**
University of Maryland, College Park

**Itai Shapira**[*]
Harvard University

## ABSTRACT

Federated learning promises significant sample-efficiency gains by pooling data across multiple agents, yet incentive misalignment is an obstacle: each update is costly to the contributor but boosts every participant. We introduce a game-theoretic framework that captures heterogeneous data: an agent's utility depends on who supplies each sample, not just how many. Agents aim to meet a PAC-style accuracy threshold at minimal personal cost. We show that uncoordinated play yields pathologies: pure equilibria may not exist, and the best equilibrium can be arbitrarily more costly than cooperation. To steer collaboration, we analyze the cost-minimizing contribution vector, prove that computing it is NP-hard, and derive a polynomial-time linear program that achieves a logarithmic approximation. Finally, pairing the LP with a simple pay-what-you-contribute rule—each agent receives a payment equal to its sample cost—yields a mechanism that is strategy-proof and, within the class of contribution-based transfers, is unique.

## 1 INTRODUCTION

Federated learning (FL) is a collaborative training framework in which multiple agents—each holding a distinct dataset—jointly optimize a global model while keeping data local. Collaboration allows agents to tap into information spread across heterogeneous records—for example, a network of hospitals pooling imaging data from distinct patient demographics to detect rare conditions sooner. Although each agent could independently train a model, collaboration offers higher accuracy or comparable performance with significantly fewer examples (Blum et al., 2017), enhancing both individual and federation-wide welfare.

However, realizing these gains hinges on incentives. Contributing model updates incurs costs in compute, bandwidth, curation effort, and privacy risk, while the global model produced by collective learning is a non-excludable public good: once trained, every agent benefits from its accuracy regardless of individual effort. This asymmetry invites a classic free-rider dilemma (Yang et al., 2015; Ahmed & Choi, 2023; Karimireddy et al., 2022): as one agent's data lifts others' accuracy while the contributor alone incurs the cost, each participant is tempted to trim its share once its own target is met. The resulting free-riding slows training and can ultimately exhaust the data pool that makes FL viable.

Consequently, federated learning can be viewed as a strategic game: each agent picks a contribution level to maximize a private utility that trades its own labeling cost against the benefit of the joint model. Existing models of incentives in federated learning either assume each agent's utility depends on the local labeled data distributions of all agents (Blum et al., 2021), or treat agents as homogeneous, which crucially implies that data are *exchangeable* and each agent's utility depends solely on the total amount of data contributed across all agents (Karimireddy et al., 2022; Murhekar et al., 2024).

However, in typical federated learning scenarios, agents face *heterogeneous data distributions* and are primarily interested in improving performance on their own local distribution. This phenomenon—local data typically yielding greater marginal utility for local performance than data

---

[*]Authors listed alphabetically. Correspondence to: Itai Shapira `itaishapira@g.harvard.edu`. Han Shao: Work mostly done while at Harvard University.

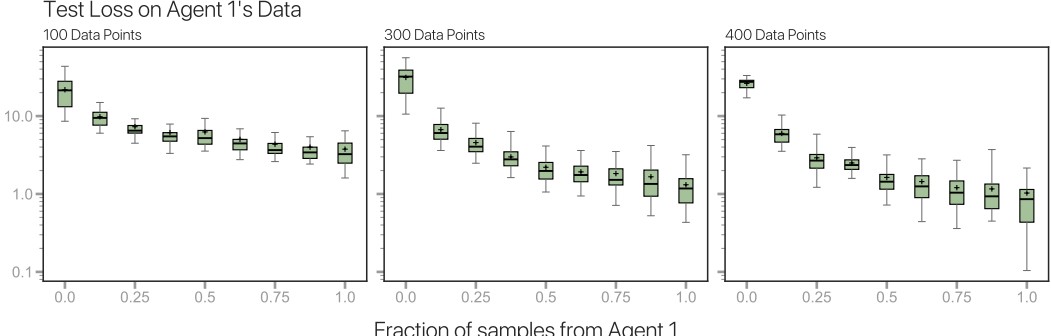

Figure 1: An agent's expected loss falls as a larger share of a fixed training set comes from their *own* distribution. With a budget of $m$ data points, we sample $\lambda m$ from Agent 1 and $(1-\lambda)m$ from Agent 2 on FEMNIST (Caldas et al., 2018), train a classifier, and repeat 100 times. For each $m$, Agent 1's loss decreases monotonically in $\lambda$, confirming that data are *not* exchangeable—utilities depend on *who* contributes. Full details in Appendix A.1.

from other agents—has been well documented in the personalization and domain adaptation literature (Ben-David et al., 2006; Bhunia et al., 2021; Hsu et al., 2019), and we empirically confirm that it is present in federated learning, as shown in Figure 1. We therefore model utilities that depend on *who* supplies each sample, not just how many.

In this work we ask: *how can incentives be aligned in this heterogeneous game?* We adopt a PAC accuracy objective—each agent wants their test error below $\varepsilon$ with confidence $1 - \delta$—and study the game induced by this objective. Left on their own, agents settle into a contribution equilibrium that can be inefficient: some agents free-ride, others overspend, and total cost can explode relative to full cooperation. The central challenge is therefore to design contribution and transfer rules that coax self-interested, heterogeneously distributed agents into pooling enough data so that each meets its own accuracy target on its own distribution, while keeping the federation's total cost near minimal.

**Our Results.** Section 2 builds a data-heterogeneous FL game with PAC thresholds: an agent's utility depends on *who* supplies the samples, not just how many. Section 3 shows that decentralized play can fail badly—pure Nash equilibria may not exist and, when they do, their total cost can exceed the cooperative optimum by an unbounded factor (Price of Stability $\to \infty$). Motivated by this gap, Section 4 assumes a planner with full information and full control. We prove that computing the cost-minimizing contribution vector is NP-hard (Theorem 2). Nonetheless, the PAC constraints admit a linear program (LP) whose cost is within a logarithmic factor of the optimum (Theorem 3); we use this LP allocation as the foundation of the mechanism design that follows. Section 5 drops the full-information assumption: agents now report private distributions. Pairing the LP allocation with a simple *pay-what-you-contribute* rule—each agent receives a payment equal to its sample cost—yields a mechanism that is strategyproof, and we derive conditions under which it uniquely satisfies these properties. Finally, while our main contributions are theoretical, we complement them with empirical validation and illustrative simulations in Appendix A that demonstrate the practical relevance of our assumptions and algorithms.

**Related Work.** Early work in FL centered on communication-efficient optimization and fairness under fully cooperative, i.i.d. or average-loss assumptions, with no pricing of data or modeling of strategic behavior. For instance, FedAvg optimizes average loss, while Agnostic FL and q-FFL reweight loss for worst-case or fairness gains, all assuming truthful reports (McMahan et al., 2017; Mohri et al., 2019; Li et al., 2020). Blum et al. (2017) introduced a collaborative PAC model and showed that pooling across $k$ heterogeneous tasks cuts sample complexity by $\widetilde{O}(\log k)$, but participation is mandatory and incentives are ignored.

Subsequent work introduced incentives but frequently assumed that agents' data are exchangeable, so that model accuracy depends only on the total sample count contributed across all participants. Under this framework, various incentive strategies—such as transfer payments or reputational re-

wards—have been proposed to combat free-riding. For example, Karimireddy et al. (2022) tie each agent's model quality to its data contribution (no external payments), whereas Murhekar et al. (2024) design budget-balanced monetary transfers that implement welfare-maximizing equilibria; and (Lin et al., 2019; Kang et al., 2019; Sarikaya & Ercetin, 2019; Ding et al., 2020; Fraboni et al., 2021) rely on reputation mechanisms and credit sharing. These designs treat data as interchangeable: model quality depends solely on the total sample count, so the marginal value of each sample ignores *who* provides it, overlooking realistic heterogeneity across data distributions.

A separate line of work studies heterogeneous data, aiming to capture how collaboration might form among agents with distinct interests. Donahue & Kleinberg (2021), and Hasan (2021) analyze coalition and Nash stability in model-sharing games and show that agents split into sub-coalitions when a global model biases some distributions. Blum et al. (2021) further show that in a personalized PAC game with distribution-specific payoffs, pure Nash equilibria may not exist, and when they do, they can be arbitrarily inefficient, underscoring the fragility of cooperation absent incentives. These game-theoretic models emphasize that heterogeneity can severely complicate collaboration: individual incentives may fail to align with socially optimal pooling of data. Our work extends these papers by introducing a concrete utility model based on PAC-style threshold guarantees where each agent requires that the global model meet a distribution-specific accuracy threshold with high confidence.

Recent studies tackle truthfulness under heterogeneity: peer-prediction payments (Pang et al., 2022), budget-balanced truthful gradient schemes (Chakarov et al., 2024), and an optimal truthful mechanism for data sharing with interdependent valuations (Chen et al., 2022); yet none compute minimum-cost allocations that meet each agent's welfare target.

## 2 MODEL

### 2.1 SETUP AND LEARNING PROTOCOL

We now make the collaborative game precise. Each agent selects how many examples to contribute; the federation pools these examples, trains a model, and each agent then evaluates the model on its own data distribution.

**Agents and data.** Consider $k$ agents $\mathcal{A} = \{1, \ldots, k\}$ who wish to jointly learn a shared predictor but individually decide how much data to contribute. Let $\mathcal{X}$ denote the instance space and $\mathcal{Y}$ the label space. A hypothesis is a function $h : \mathcal{X} \to \mathcal{Y}$ mapping instances to predicted labels. Fix a hypothesis class $\mathcal{H}$ with VC dimension $d$, and assume agents seek to approximate an unknown target function $h^\star \in \mathcal{H}$. Each agent $i \in \mathcal{A}$ has access to a local marginal data distribution $\mathcal{D}_i$ over $\mathcal{X}$ and can query labels for any data point drawn from this distribution. The collaborative learning process involves two stages:

**Stage 1 – Sample Collection.** Each agent $i$ chooses a contribution level $m_i \in \mathbb{N}$, draws an i.i.d. *unlabeled* dataset $U_i \sim \mathcal{D}_i^{m_i}$, and queries the true labels of these samples, which are determined by the target function $h^\star$. The labeled sets are pooled into a dataset

$$S = \bigcup_{i \in \mathcal{A}} \{ (x, h^\star(x)) \mid x \in U_i \}.$$

Denote by $\mathcal{P}(\boldsymbol{\mathcal{D}}, \mathbf{m}, h^\star)$ the distribution of this dataset, which is determined by the marginal data distributions $\boldsymbol{\mathcal{D}} = (\mathcal{D}_1, \ldots, \mathcal{D}_k)$, the contribution profile $\mathbf{m} = (m_1, \ldots, m_k)$, and $h^\star$.

**Stage 2 – Model Training.** A central server trains a model using Empirical Risk Minimization (ERM) method:

$$\text{ERM}(S) = \{h \in \mathcal{H} \mid \text{err}_S(h) = \min_{h' \in \mathcal{H}} \text{err}_S(h')\},$$

where $\mathrm{err}_S(h)$ is the empirical error of hypothesis $h$ on dataset $S$.[1] For any hypothesis $h$, marginal data distribution $\mathcal{D}$, and target function $h^\star$, the generalization error is defined as

$$\mathrm{err}_{\mathcal{D},h^\star}(h) = \mathbb{P}_{x\sim\mathcal{D}}\left(h(x) \neq h^\star(x)\right).$$

Given a labeled training set $S$, we define the generalization error of running ERM over $S$ relative to $(\mathcal{D}, h^\star)$ as:

$$\mathrm{err}_{\mathcal{D},h^\star}^{\mathrm{ERM}}(S) = \max_{h\in\mathrm{ERM}(S)} \mathrm{err}_{\mathcal{D},h^\star}(h), \tag{1}$$

to reflect the generalization performance of an empirically optimal hypothesis under distribution $\mathcal{D}$ with labeling function $h^\star$. That is, if the ERM returns multiple minimizers, we take the worst generalization error, ensuring every bound holds under arbitrary tie-breaking.

Stages 1 and 2 define our collaborative game. Each agent $i$ chooses an integer contribution level $m_i$. The server draws $m_i$ i.i.d. samples from $\mathcal{D}_i$ for each agent, pools the labeled data into $S$, and trains a single global ERM predictor $\widehat{h}(S)$ over $\mathcal{H}$. The next subsection, Section 2.2, specifies the payoffs from this global model and the cost of contributing data.

## 2.2 PAC ACCURACY OBJECTIVE

Each player ultimately wants a model whose test error on its *own* distribution is low; if collaboration fails, the fallback is to train alone. We focus on a single, widely-used performance criterion to formalize this goal: a Probably Approximately Correct (PAC) accuracy threshold.[2] In standard realizable PAC learning, an agent wants—with probability at least $(1-\delta)$—to keep its generalization error below a tolerance $\varepsilon$.[3] Here we adapt that notion to our federated setting, where (1) the learner uses ERM on the pooled samples and (2) each agent $i$ draws from its own fixed marginal distribution $\mathcal{D}_i$. We then define the PAC accuracy objective as follows: agent $i$ requires that, *for every* target hypothesis $h^\star \in \mathcal{H}$,

$$\Pr_{S\sim\mathcal{P}(\boldsymbol{\mathcal{D}},\boldsymbol{m},h^\star)}\left[\mathrm{err}_{\mathcal{D}_i,h^\star}^{\mathrm{ERM}}(S) \leq \varepsilon\right] \geq 1 - \delta. \tag{2}$$

We use $a_i^{\varepsilon,\delta}(\boldsymbol{m})$ to denote a *binary* variable that is 1 if Equation 2 is satisfied, and 0 otherwise; we refer to this as agent $i$'s $(\varepsilon, \delta)$-*requirement* or *target*.

**Cost of Contributions.** Contributing data incurs costs. Computation, and privacy risk all translate into a per-sample monetary burden. We capture this with a linear cost: when agent $i$ supplies $m_i$ samples, it pays $c_i m_i$, where $c_i > 0$ denotes the cost per sample. The parameter $c_i$ can reflect labeling fees, extra compute, or other participation overhead.

**Utility Functions.** Each agent balances the model's benefit against the data cost. We normalize monetary units so that achieving the PAC goal is worth exactly one unit of payoff, yielding a simplified expression for utility:

$$u_i(\mathbf{m}) = a_i^{\varepsilon,\delta}(\boldsymbol{m}) - c_i m_i. \tag{3}$$

To ensure every agent's goal is attainable and worth pursuing, we assume *self-sufficiency:* every agent could, in the worst case, meet its own accuracy requirement by training on its own data alone. Concretely, let $n_i^{\mathrm{ind}}$ denote the smallest number of samples that agent $i$ would need to label by itself (with no contributions from others) to satisfy the $(\varepsilon, \delta)$-requirement on $\mathcal{D}_i$. Under PAC accuracy, for all marginal distributions, it holds that

$$n_i^{\mathrm{ind}} \leq O\left(\frac{d+\ln(1/\delta)}{\varepsilon}\right),$$

where $d = \mathrm{VCdim}(\mathcal{H})$, the VC dimension of $\mathcal{H}$.

---

[1]In practice, algorithms such as FedAvg (McMahan et al., 2017) only approximate the pooled-ERM solution through local updates and model aggregation, often without accessing raw examples, which remain on users' devices and are never aggregated on the server. In our work, the optimal ERM solution is used as an upper-bound reference point, which lets us abstract away optimization dynamics and focus on incentives and sample complexity. See Section 6 for more discussion.

[2]Other metrics, such as expected error, can be treated analogously; see Section 6.

[3]The heterogeneous (different $\varepsilon_i$ for each agent) case is analogous, as we discuss in Section 6; we present the common-$\varepsilon$ case for clarity.

**Assumption 1.** $c_i \, n_i^{ind} \; < \; 1$, *for every $i$.*

This self-sufficiency implies that failing to meet the accuracy threshold results in strictly lower utility than meeting it (since an agent can always fall back on solo training to obtain the benefit, albeit at potentially high cost). Therefore, the agent-level decision problem in Equation 3 is to choose $m_i$ so as to minimize its own cost $c_i m_i$ subject to satisfying its $(\varepsilon, \delta)$-accuracy requirement in Equation 2.

This binary threshold payoff can represent a broad class of quasi-linear utility functions, and captures richer behavior than its simple indicator form suggests at first glance. Under regularity assumptions, Appendix B shows via a Lagrange-multiplier argument that any optimal solution to above constrained problem is also optimal for some unconstrained objective that trades off continuous accuracy against cost; in particular, there is a multiplier $\lambda_i$ such that the optimal contribution $m_i^*$ that meets the PAC constraint also maximizes a quasi-linear utility that penalizes cost at rate $\lambda_i$. Intuitively, $\lambda_i$ is the agent's shadow price for accuracy, and the point where our indicator flips from $0$ to $1$ is exactly where the marginal benefit of additional accuracy falls to this price. Thus the threshold model is a concise way to capture the saturation behavior of a wide family of smooth, monotone, diminishing-returns utilities, rather than an assumption that agents literally have zero value below the target.

**Social Cost Optimization.** A central planner seeks to maximize *utilitarian social welfare*, namely $\sum_{i \in \mathcal{A}} u_i(\mathbf{m})$. Under Assumption 1, this welfare objective coincides with minimizing the aggregate labeling cost subject to the same PAC constraints. Thus, from a global standpoint, the problem is to find the contribution profile that lets every agent meet its accuracy target at the lowest possible total cost.

$$\min_{\mathbf{m}} \; \mathbf{c}^\top \mathbf{m} \qquad\qquad (4)$$
$$\text{s.t.} \quad a_i^{\varepsilon,\delta}(\boldsymbol{m}) = 1 \quad \forall \, i \in \mathcal{A}.$$

Here $\mathbf{c}^\top \mathbf{m}$ is the federation's total labeling cost, and the constraint is the PAC guarantee of Equation 2: for every agent $i$ and every target hypothesis $h^\star$, the ERM model must, with probability at least $1 - \delta$, achieve error at most $\varepsilon$ on $\mathcal{D}_i$. The planner therefore seeks the cheapest contribution profile that satisfies *all* agents' $(\varepsilon, \delta)$-requirements simultaneously.

**Warm-Up Example.** Consider a single agent whose marginal distribution is uniform over the $n$ distinct points $x_1, \ldots, x_n \subset X$ and a hypothesis class that contains every possible labeling of those points. When $\varepsilon < 1/n$, satisfying the PAC accuracy threshold below $\varepsilon$ with probability at least $1 - \delta$ requires drawing enough samples (with replacement) so that every point appears at least once. With $k > 1$ agents, we can see how collaboration can improve sample complexity: consider $k$ agents whose distribution places most of its probability mass on a different point $x_i$, while still assigning every other point probability at least $\varepsilon$. By pooling data, agents supply one another with examples they rarely see, allowing each to collect fewer local samples while the group still satisfies the PAC threshold.

## 3 THE INEFFICIENCY OF EQUILIBRIA

With the cooperative optimum in hand (Equation 4), we now analyze the uncoordinated game in which each agent strategically chooses its sample size. We quantify the cost gap between equilibrium behavior and the social optimum and show that a Nash equilibrium (NE) can be arbitrarily more costly. This gap motivates central coordination, which we develop next, and it serves as the conceptual starting point for the rest of the paper.

A contribution profile $\mathbf{m}$ is a NE if no agent can raise its utility by unilaterally changing its sample count. Since each agent can always fall back to its self-sufficient plan $n_i^{\text{ind}}$, an equilibrium exists only when every accuracy threshold is met; otherwise the under-served agent would deviate to $(n_i^{\text{ind}}, \mathbf{m}_{-i})$ to raise its utility. Thus, at equilibrium no agent wants to cut its contribution below the threshold or add extra samples.

It is straightforward to show that pure NE may not exist in our framework (Theorem 6), whereas mixed NE always exists (Theorem 5). Even when a pure equilibrium exists, it can be highly

inefficient. To illustrate this, consider a simple two-player setting. Let the instance space be $\mathcal{X} = \{x_A, x_B\}$. Two players, Alice and Bob, each place most of their probability mass on a different point: for a small $\varepsilon \in (0, \frac{1}{4})$,

$$\mathcal{D}_{\text{Alice}}(x_A) = 1 - 2\varepsilon, \qquad\qquad \mathcal{D}_{\text{Alice}}(x_B) = 2\varepsilon,$$

$$\mathcal{D}_{\text{Bob}}(x_A) = 2\varepsilon, \qquad\qquad \mathcal{D}_{\text{Bob}}(x_B) = 1 - 2\varepsilon.$$

Under the PAC objective, each agent requires that *both* points be correctly classified with high probability. If Alice and Bob each contribute one labeled example, then with probability $(1 - 2\varepsilon)^2$ Alice draws $x_A$ while Bob draws $x_B$. Thus $(m_{\text{Alice}}, m_{\text{Bob}}) = (1, 1)$ meets both thresholds at total cost $c_{\text{Alice}} + c_{\text{Bob}} = \Theta(1)$ and is optimal. If one free-rides, the other must keep sampling until seeing *both* points, requiring $\Omega(1/\varepsilon)$ samples. Since neither player can unilaterally lower this cost, this profile is an NE with a cost of $\Omega(1/\varepsilon)$.

This example shows that some equilibria can be much costlier than the social optimum even though a cost-minimal equilibrium exists. How large can the best equilibrium's cost be? We formalize it with the Price of Stability (PoS) (Anshelevich et al., 2008): the ratio of total samples in the least-cost equilibrium to those in the planner's optimum. PoS captures the cost of decentralization. Even the best self-enforcing outcome can require far more samples than the cooperative minimum. In the following, we show that PoS can be arbitrarily large:

**Theorem 1** (PoS is unbounded). *For any $\varepsilon \in (0, 1)$ there exists a sequence of instances of our problem in which the ratio of the best NE to the optimal solution approaches $\Omega\big(\log(1/\varepsilon)\big)$.*

## 4 CENTRAL COORDINATION WITH FULL INFORMATION

The unbounded price of stability in Section 3 shows that self-directed contribution games may squander resources. A natural fix is to appoint a *central planner*, such as a regulator or platform operator, who can dictate how many labeled examples each agent contributes. Before addressing strategic issues, we begin with the technical question of whether the planner can compute the cost-minimizing contribution vector efficiently. In this section we show that the problem is NP-hard (Theorem 2), yet it admits an efficient approximation via an LP with logarithmic-type guarantees (Theorem 3). The next section returns to strategy under coordination and builds on this to design payments that align incentives.

To isolate the computational question and remove strategic frictions for now, in this section we grant the planner two powers. First, it has *full information*—it observes every marginal distribution $\mathcal{D}_i$ and each per-sample cost $c_i$. Second, it has *complete control*—if an agent opts in, the planner can compel it to supply the prescribed number of samples $m_i$. These assumptions fit cases where data statistics are public—e.g., hospital demographics or published mobile-traffic summaries—and participation is contractual. With strategic frictions removed, the task is now purely computational: who should collect how many samples? We tackle that question here and relax the full-information assumption in Section 5.

### 4.1 COLLABORATIVE PAC SAMPLE-ALLOCATION PROBLEM

Given a contribution vector $\mathbf{m}$, let $S$ be the pooled sample obtained by drawing $m_i$ points independently from each $\mathcal{D}_i$. We call $\mathbf{m}$ *feasible* if, with probability at least $1 - \delta$, every ERM hypothesis trained on $S$ has error at most $\varepsilon$ on every $\mathcal{D}_i$. The planner specifies only the draw counts from each distribution. Under these full-information assumptions, the task reduces to Equation 4. Our first question is thus computational: is this optimization tractable? We answer in the negative. The result below shows NP-hardness in $|\mathcal{H}|$ even with a single agent. The full proof is in Appendix D.1.

**Theorem 2.** *Under the PAC-accuracy objective, determining whether a specified sample count $m$ suffices to meet the $(\varepsilon, \delta)$-requirement is NP-hard with respect to the hypothesis-class size $|\mathcal{H}|$, even when there is only one agent.*

### 4.2 APPROXIMATION VIA LINEAR PROGRAMMING

Despite this hardness barrier, the structure of the PAC constraints admits an efficient relaxation. Solving this LP takes polynomial time and returns a contribution vector whose total cost is within a

logarithmic factor of the optimum. More specifically, for a finite class $\mathcal{H}$, our task is to find a vector of sample counts $\mathbf{m} = (m_i)_{i \in \mathcal{A}}$ that—with probability at least $1 - \delta$—forces ERM to discard every hypothesis $h$ that is *bad* for some agent:

$$\exists\, i \in \mathcal{A}: \ \mathcal{D}_i\big(\{\, x : h(x) \neq h^\star(x) \,\}\big) \ \geq \ \varepsilon.$$

For any ordered pair $(h_1, h_2)$ satisfying this condition, the probability that *no* sample lands in the set $\{x : h_1(x) \neq h_2(x)\}$ is $\prod_{i \in \mathcal{A}} \big(1 - \mathcal{D}_i(\{h_1 \neq h_2\})\big)^{m_i}$, which is log-linear in $\mathbf{m}$. We can therefore convert Equation 4 into an LP with polynomially many constraints, which can be solved efficiently.

For infinite $\mathcal{H}$, we provide an approximate solution by solving the LP over a finite cover $\overline{\mathcal{H}} \subset \mathcal{H}$, whose size is polynomial in $\frac{1}{\varepsilon}$ and $\frac{1}{\delta}$ when the VC dimension $d = \mathrm{VCdim}(\mathcal{H})$ is bounded. While the resulting solution ensures PAC constraints are satisfied for $\overline{\mathcal{H}}$, it may not suffice for the full class $\mathcal{H}$, as there may exist a hypothesis $h \in \mathcal{H} \setminus \overline{\mathcal{H}}$ that is consistent but still incurs high error. Nevertheless, we show that scaling the solution by a factor of roughly $d$ suffices to ensure the PAC objective is met for all of $\mathcal{H}$.

Any server that implements this LP to coordinate contributions, as in Section 5, interacts with agents only through the information the LP requires, and two practical aspects are worth noting. First, the LP needs from each agent only a finite, explicitly specified vector probabilities, so the communication burden is well defined and finite.[4] Second, these quantities are summary distributional statistics that each agent can compute locally from its own data. Sharing only these statistics, rather than raw examples, ensures the server never receives individual training examples.

Formally, the next theorem summarizes the approximation guarantee of this LP relaxation; the full proof appears in Appendix D.2. As an empirical check, Appendix A.2 validates this LP allocation on a finite hypothesis class by comparing to the true optimum $\sum_i m_i^\star$ across varying $|\mathcal{H}|$.

**Theorem 3** (Approximation via Linear Programming). *Given any $\mathcal{H}$ and $\varepsilon, \delta > 0$:*

- *For finite $\mathcal{H}$, the LP over $(\mathcal{H}, \varepsilon, \delta)$ returns a $\frac{\log(1/\delta) + \log|\mathcal{H}|}{\log(1/\delta)}$-approximate solution to Equation 4.*

- *For infinite $\mathcal{H}$, running the LP over $(\overline{\mathcal{H}}, \varepsilon, \delta')$ and multiplying it by $d + \log(1/\delta'')$ returns a $O\big(\frac{d^2 (\log(c_{max} k d / (c_{min} \varepsilon \delta)))^2}{\log(1/\delta)}\big)$-approximate solution to Equation 4, where $\overline{\mathcal{H}} \subset \mathcal{H}$ is a $\gamma$-cover of $\mathcal{H}$, $\gamma = \Theta\big(c_{min} \varepsilon \delta / (c_{max} k (d + \log(1/\delta)))\big)$, $d = \mathrm{VCdim}(\mathcal{H})$, $\delta'' = \frac{\delta}{4|\overline{\mathcal{H}}|}$ and $\delta' = \frac{\delta}{8(d + \log(2|\overline{\mathcal{H}}|/\delta))}$, $c_{min} = \min_{i \in [k]} c_i$ and $c_{max} = \max_{i \in [k]} c_i$.*

## 5 MECHANISM DESIGN WITH APPROXIMATE SOLUTIONS

In Section 4 we studied the planner's computational problem under full information and control. Here we keep prescriptive control but drop full information and reintroduce strategy under coordination. In the uncoordinated game of Section 3 agents act by choosing their own sample sizes. With the planner fixing sample counts, agents can only influence outcomes by misreporting their local distributions, which shifts the LP constraints from Section 4, and thus the computed contribution vector.[5] Building on that LP, in this section, we design a mechanism that is strategyproof and unique within a broad contribution-based class, blocking this manipulation and addressing the inefficiency in Section 3. Together, these results highlight the contrast between uncoordinated and coordinated settings.

---

[4]Formally, for each pair $h_1, h_2 \in \mathcal{H}$ the planner needs, from each agent $i$, the probability that its data lie in the disagreement set $\{x : h_1(x) \neq h_2(x)\}$, that is, the quantity $\mathcal{D}_i(\{x : h_1(x) \neq h_2(x)\})$. If $\mathcal{H}$ is finite, this yields at most $|\mathcal{H}|^2$ numbers per agent. If $\mathcal{H}$ is infinite, Appendix D.2 replaces $\mathcal{H}$ with a finite $\gamma$-cover $\overline{\mathcal{H}}$ whose size is polynomial in $d = \mathrm{VCdim}(\mathcal{H})$. These are summary statistics that can be estimated locally or on a shared unlabeled validation set, rather than raw examples.

[5]Enforcing prescribed contributions without accessing raw data raises a verification challenge. Practical approaches include cryptographic proofs and auditable contribution records (Ma et al., 2024; Kumar et al., 2025).

Specifically, we now regard the agents' local marginal data distributions as *private information* that must be reported to the mechanism. Their per-sample costs $c_i$ remain known parameters.[6] To alleviate the resulting incentive issues, we allow payments to the agents. The sequence of events, therefore, is as follows:

1. Agents report their local marginal data distributions, $\boldsymbol{\mathcal{D}^r} = (\mathcal{D}_1^r, \ldots, \mathcal{D}_k^r)$.
2. The central planner computes a solution $\mathbf{m}$ based on the reported distributions $\boldsymbol{\mathcal{D}^r}$.
3. Agents contribute according to $\mathbf{m}$.
4. The central planner pays each agent $i$ an amount $p_i$.

We make the standard assumption that agent utilities are quasi-linear, that is, the utility of agent $i$ for contribution vector $\mathbf{m}$ and payment $p_i$ (from the mechanism to the agent) is $u_i(\mathbf{m}) + p_i$. We say that a mechanism (which computes agent contributions and payments) is *strategyproof* if an agent can never benefit from misreporting their local marginal distribution; in game-theoretic terms, it is a dominant strategy to report $\mathcal{D}_i^r = \mathcal{D}_i$.

The challenge now is twofold: computation and incentives. In other words, the question is this: *How can we* tractably *compute contributions and payments such that the resulting mechanism is* strategyproof?

## 5.1 THE PAY-WHAT-YOU-CONTRIBUTE MECHANISM

In our model, the answer to the foregoing question is surprisingly immediate. To compute the contribution vector $\mathbf{m}$, we use the approximation algorithm of Theorem 3. For payments, we use the simple *pay-what-you-contribute (PWYC)* scheme, that is, compensate each agent for its contribution, up to a constant:

$$p_i(\mathbf{m}) = c_i \cdot m_i + C_i, \tag{5}$$

for constants $C_1, \ldots, C_k$.

Why is PWYC strategyproof? The utility of agent $i$ when reporting truthfully is

$$u_i(\mathbf{m}) = a_i^{\varepsilon,\delta}(\boldsymbol{m}) - c_i \cdot m_i + c_i \cdot m_i + C_i = 1 + C_i,$$

as its learning threshold is satisfied by the contribution vector $\mathbf{m}$ computed by the approximation algorithm. Note that this is the maximum possible utility under this mechanism: for any $\mathbf{m}'$, the utility of agent $i$ is either $1 + C_i$ or $C_i$, depending on whether its learning threshold is satisfied by $\mathbf{m}'$.

## 5.2 ALTERNATIVES TO PAY WHAT YOU CONTRIBUTE?

While the PWYC mechanism is strategyproof, one may wonder whether it is possible to design more sophisticated mechanisms with the goal of, for example, minimizing payments. A natural candidate is the classic Vickrey-Clarke-Groves (VCG) mechanism, which in our setting computes the optimal solution $\mathbf{m}^{\text{OPT}}$, asks agents to contribute according to $\mathbf{m}^{\text{OPT}}$, and then pays each agent $i$

$$p_i = \sum_{j \neq i} u_j(\mathbf{m}^{\text{OPT}}) + q_i(\boldsymbol{\mathcal{D}}_{-i}) = k - 1 - \sum_{j \neq i} c_j \cdot m_j^{\text{OPT}} + q_i(\boldsymbol{\mathcal{D}}_{-i}^r), \tag{6}$$

where $q_i(\boldsymbol{\mathcal{D}}_{-i})$ is a term independent of the report $\mathcal{D}_i$ of agent $i$. By standard arguments (Nisan, 2007), the VCG mechanism ensures that each agent reports truthfully.

An obstacle to using VCG directly is that computing the optimal contribution vector is computationally hard (Theorem 2). In the *algorithmic mechanism design* (Nisan & Ronen, 2001) literature, however, there are various mechanisms that overcome computational hardness by augmenting approximation algorithms with clever payment schemes, including ones inspired by VCG (Lehmann et al., 2002; Dobzinski, 2007). Is there such a rich mechanism design space in our setting? We give a

---

[6]For many applications, it is natural to treat per-sample costs as known or contractually specified: platform may post a fixed price per labeled example, meter and bill bandwidth or compute usage, or sign annotation contracts that set a per-label rate up front, so that $c_i$ is a parameter of the environment rather than private strategic information.

partial negative answer to this question, showing that the PWYC mechanism is, in a qualified sense, unique.

This uniqueness is unfortunate, since the absence of a budget-balanced mechanism naturally raises the concern that shifting costs to the planner could, in a sense, diminish some of the benefits of FL training in the first place. We view the uniqueness of PWYC as a substantive, if somewhat negative, insight: if one insists on dominant-strategy truthfulness and approximate efficiency in a heterogeneous FL game, then some external budget or cost sharing is unavoidable, much like in other public-goods settings where reimbursement is often needed to align incentives. Even in standard mechanism design settings, including the celebrated VCG mechanism, exact budget balance is generally impossible, so the planner must make net payments to align incentives. Importantly, PWYC does not force the planner to shoulder all costs: the constants $C_i$ in Equation 5 can be set to recover part of the spend from participants or downstream users while preserving strategyproofness.

We start by defining a class of approximate algorithms and a class of "easy-to-compute" payment rules. We first introduce a *local obliviousness* property of contribution solutions, which helps distinguish approximate solutions from exact optima.

**Definition 1** (Locally Oblivious Approximations). *Given an approximation algorithm APPROX and a solution* $\mathbf{m}$*, we say that the algorithm is* locally oblivious *at* $\mathbf{m}$ *if, for any neighbor* $\mathbf{m}'$ *with* $\|\mathbf{m}' - \mathbf{m}\|_1 = 1$*, there exist distributions* $\boldsymbol{\mathcal{D}} = (\mathcal{D}_1, \ldots, \mathcal{D}_k)$ *and* $\boldsymbol{\mathcal{D}}' = (\mathcal{D}_1', \ldots, \mathcal{D}_k')$ *such that:*

- $\mathbf{m} = APPROX(\boldsymbol{\mathcal{D}})$ *and* $\mathbf{m}' = APPROX(\mathcal{D}_i', \boldsymbol{\mathcal{D}}_{-i})$ *for all* $i \in [k]$,

- *Both* $\mathbf{m}$ *and* $\mathbf{m}'$ *are feasible for* $\boldsymbol{\mathcal{D}}$ *and for each* $(\mathcal{D}_i', \boldsymbol{\mathcal{D}}_{-i})$*, for all* $i \in [k]$.

Local obliviousness highlights a key difference between approximate solutions and exact optima. Intuitively, it reflects a certain slack in approximation: for any solution $\mathbf{m}$ and any neighbor, we can construct pairs of distribution vectors where the approximation algorithm outputs $\mathbf{m}$ and its neighbor respectively, and both solutions remain feasible across all these instances. For example, we show that the approximation algorithm introduced in Theorem 3 is locally oblivious at all $\mathbf{m}$ satisfying $m_1, m_2 \geq 2|\mathcal{H}| \log |\mathcal{H}|$ in the two-agent setting.

**Lemma 1.** *For any $H$ greater than a universal constant $C$, there exists a PAC learning instance of $(\mathcal{H}, \varepsilon, \delta)$ in the two-agent setting with $|\mathcal{H}| = H$ such that the approximation algorithm introduced in Theorem 3 is oblivious at all $\mathbf{m}$ with $m_1, m_2 \geq 2|\mathcal{H}| \log |\mathcal{H}|$.*

In general, a payment mechanism maps the reported distributions $\boldsymbol{\mathcal{D}}^r$ to a payment vector $\boldsymbol{p}$. However, it is often unclear how to effectively utilize the reported distributions directly. To address this, we consider a class of mechanisms that are easy to compute in practice—those that depend on the distributions only through the resulting contribution solution $\mathbf{m}$.

**Definition 2** (Contribution-Based Mechanisms). *Given an algorithm APPROX, a payment mechanism $\boldsymbol{p}$ is* contribution-based *if there exists* $\mathbf{f} : \mathbb{N}^k \to \mathbb{R}^k$ *and* $\mathbf{q}$ *such that*

$$p_i(\boldsymbol{\mathcal{D}}^r) = f_i\big(APPROX(\boldsymbol{\mathcal{D}}^r)\big) + q_i(\boldsymbol{\mathcal{D}}_{-i}^r),$$

*Thus agent $i$'s own report affects $p_i$ only via the computed contribution vector.*

We now show that PWYC is the only strategyproof contribution-based payment mechanism when approximate solutions are used for two agents (see proof in Appendix E):

**Theorem 4.** *For any strategyproof contribution-based payment mechanism $\mathbf{f}$, approximation algorithm APPROX, and connected $M \subset \mathbb{N}^k$, if APPROX is oblivious at all $\mathbf{m} \in M$, there exist constants $C_1, C_2$ such that for all $\mathbf{m} \in M$:*

$$f_i(\mathbf{m}) = c_i \cdot m_i + C_i, \forall i = 1, 2,$$

**Corollary 1.** *For any $H$ greater than a universal constant $C$, there exists a PAC learning instance of $(\mathcal{H}, \varepsilon, \delta)$ in the two-agent setting with $|\mathcal{H}| = H$ such that when applying the approximation algorithm introduced in Theorem 3 to compute the contribution solution, the strategyproof contribution-based payment $\mathbf{f}$ must satisfy*

$$f_i(\mathbf{m}) = c_i \cdot m_i + C_i, \forall i = 1, 2,$$

*for all $\mathbf{m}$ with $m_1, m_2 \geq 2|\mathcal{H}| \log |\mathcal{H}|$ for some constants $C_1, C_2$.*

Corollary 1 follows by combining Theorem 4 with Lemma 1. Appendix F extends the same approximation to a broader class of objectives.

## 6 DISCUSSION

We conclude by discussing the implications of our results, examining key modeling assumptions, and outlining connections to broader theory.

**Beyond the PAC accuracy objective.** The $(\varepsilon, \delta)$-guarantee controls the *tail*: with probability at least $1 - \delta$ the generalization error does not exceed $\varepsilon$. A complementary member of the same threshold family bounds the *expected* error by $\varepsilon$, trading worst-case assurance for an average-case criterion that may better reflect practical risk tolerance. Most of our analysis carries over unchanged: Appendix F.1 develops approximation schemes for computing optimal contribution vectors under expected error, and Appendix F.2 proves that these solutions continue to satisfy the local obliviousness property.

Taking a broader view of utilities, we focus on *threshold payoffs*. With quasi-linear utility, maximizing utility equals minimizing cost subject to the threshold. This replaces the intractable dependence of each agent's loss on its contribution with a tractable optimization. Future work can explore richer models that trade off cost and payoff more flexibly.

**Centralized ERM as a benchmark.** In this work, the ERM solution on the pooled dataset $S$ is used only as a theoretical benchmark, not as an implementation assumption. The raison d'être of FL is to avoid aggregating raw data on a central server, so directly minimizing the joint empirical loss over all local data imagines having everything in one place. In practice, algorithms such as FedAvg approximate the pooled-ERM solution via local updates and model aggregation, without ever materializing a pooled dataset. In our analysis, this centralized ERM solution serves as the reference point: it defines the PAC constraints and certifies that, if one were to train on the pooled draw of size $m$, the resulting model would satisfy each agent's $(\varepsilon_i, \delta)$ requirement. This abstraction lets us ignore optimization dynamics and focus on incentives and sample complexity, while any FL protocol that gets sufficiently close to this benchmark inherits the same qualitative guarantees. Using centralized ERM as an upper-bound reference is standard in FL theory (Mohri et al., 2019; Li et al., 2020; Blum et al., 2021).

**Common accuracy threshold.** For clarity, we assume a common accuracy target $\varepsilon$ across agents. This simplifying assumption streamlines notation and highlights the structural properties of the contribution game. However, our results do not rely on uniformity. Each agent's objective can be parameterized by a distinct $\varepsilon_i$, and all definitions, equilibrium analyses, and approximation guarantees extend naturally to this heterogeneous setting.

**Linear costs.** Linear per-sample costs simplify analysis and are common in the literature, but they can be restrictive and not strictly necessary. Permitting any convex, non-decreasing cost function captures realistic scenarios with increasing marginal cost—for instance, later data points may be more expensive to collect. The strategic conclusions are unchanged: a NE still exists and pure equilibria need not. Meanwhile, the *linear* program in our approximation algorithm becomes a convex program that remains tractable. Thus, the incentive mechanism we derive continues to work even when agents' data-generation costs rise with effort, at the price of solving slightly more involved convex optimizations.

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

## A   EMPIRICAL VALIDATION AND ILLUSTRATIVE SIMULATIONS

**Overview.**   Our main contributions are theoretical. This appendix complements them with two empirical studies that illustrate the modeling assumptions and the practical significance of the planner and mechanism. First, we show that data are not exchangeable, since an agent benefits more from data drawn from its own distribution. Second, we validate the planner's LP allocation on a finite hypothesis class by comparing its total cost to the true optimum found by search.

**Dataset Selection and Motivation.**   We build on the LEAF benchmark (Caldas et al., 2018), which provides realistic federated datasets naturally partitioned by user. We selected two datasets, **FEMNIST** and **Shakespeare**, to cover both vision and textual tasks. These specific datasets were chosen because they have the highest average number of datapoints per user among the LEAF collection (Caldas et al., 2018). FEMNIST, built on Extended MNIST (LeCun, 1998; Cohen et al., 2017), consists of handwritten character images partitioned by writer. Shakespeare is constructed from *The Complete Works of William Shakespeare* (Shakespeare et al., 1989; McMahan et al., 2017), partitioned by speaking role, with each role in each play treated as a distinct agent.

**Dataset Construction.**   We construct two-agent subsets from each dataset by selecting the two users with the most data points. For FEMNIST, these are writers `f0261_06` and `f0289_10`, jointly contributing $1,047$ samples. For Shakespeare, we select the roles Hamlet (from *Hamlet*) and Iago (from *Othello*), together providing $112,116$ samples. Additionally, we create a variant scenario with one top agent and a random convex combination of five other agents forming the second agent. We illustrate qualitative differences between the FEMNIST agents in Figure 2.

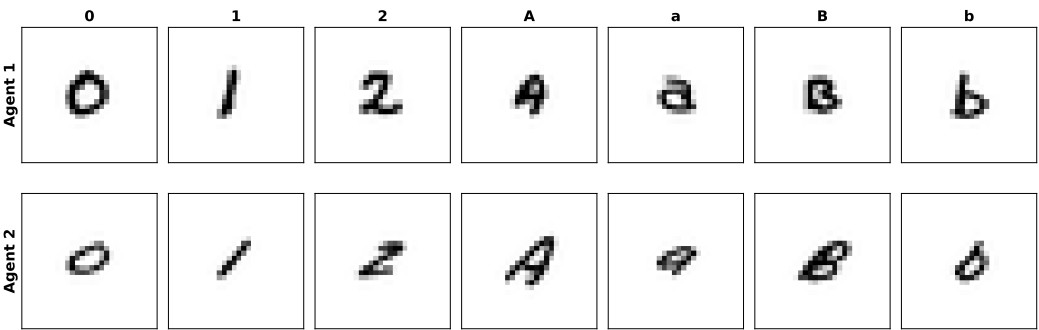

Figure 2: Samples from selected character classes for two FEMNIST agents, illustrating distinct handwriting styles.

### A.1   EFFECT OF DATA COMPOSITION ON AGENT-SPECIFIC PERFORMANCE

**Experimental Setup.**   For varying total data size $m$ and fraction $\lambda \in [0, 1]$ from Agent 1, we construct datasets $D^{(m,\lambda)}$ comprising $\lambda m$ samples from Agent 1 and $(1 - \lambda)m$ samples from Agent 2. Models trained on these datasets are evaluated separately on each agent's test set. We use standard cross-entropy loss and the SOAP optimizer (Vyas et al., 2024). To capture performance variability, experiments are repeated 100 times with different random seeds. See Table 1 for detailed parameters. We employed task-appropriate model architectures aligned with prior federated learning works (McMahan et al., 2017; Caldas et al., 2018; Murhekar et al., 2024). For FEMNIST (image classification), the model is a lightweight CNN. For Shakespeare (next-character prediction), we use a recurrent neural network with an LSTM backbone.

Table 1: Summary of experimental parameters.

| Dataset | Agent 1 | Agent 2 | # Seeds | $m$ values | Learning rate | Epochs |
|---------|---------|---------|---------|-----------|---------------|--------|
| FEMNIST | `f0261_06` | `f0289_10` | 100 | $10 - 450$ | 0.01 | 50 |
| Shakespeare | Hamlet | Iago | 100 | $10,000 - 45,000$ | 0.01 | 25 |

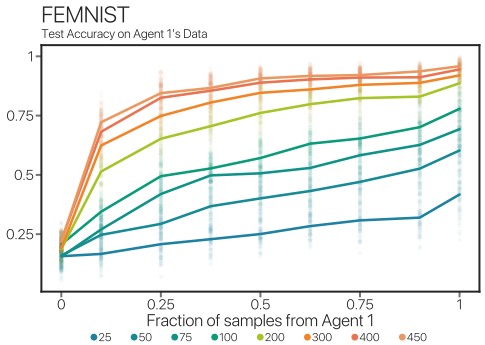 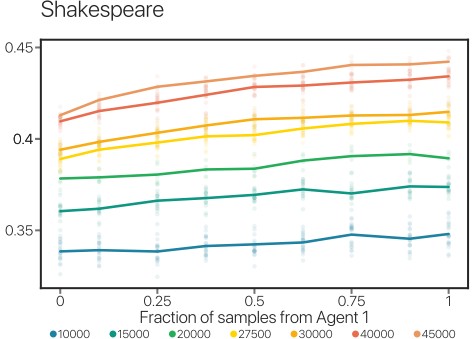

Figure 3: Agent 1's test accuracy as a function of the fraction of samples contributed from its own distribution (Agent 1), compared against random convex combinations from other agents (Agent 2). Each curve represents different total dataset sizes ($m$). The monotonic trend supports the weak–monotonicity Assumption 2.

**Results and Discussion.** These experiments confirm the intuitive expectations about data volume and personalization, and quantify their effects. (i) For any fixed share $\lambda$, increasing the budget $m$ lowers both agents' losses—more data helps everyone (Assumption 2). (ii) For any fixed $m$, raising $\lambda$ cuts Agent 1's loss: when the training set includes a higher proportion of one writer's style, the model is better tuned to that style and thus generalizes better on that agent's test set. These trends mirror prior personalization findings in federated learning and domain adaptation (Hsu et al., 2019).

### A.2 PLANNER VALIDATION ON A FINITE HYPOTHESIS CLASS

We empirically validate the planner's LP allocation from Section 4.2 by comparing its total pre-scribed samples to the true optimum obtained by discrete search over integer allocations. We report the ratio

$$ r = \frac{\sum_i m_i^{\mathrm{LP}}}{\sum_i m_i^{\star}} $$

under uniform per-sample costs $c_i = 1$.

**Finite-class construction.** We focus on FEMNIST and construct a finite hypothesis class $\mathcal{H}$ by sampling checkpoints from a compact CNN with two convolutional blocks and a linear classifier. For each agent, we estimate pairwise disagreements $\mathcal{D}_i(\{h_1 \neq h_2\})$ for all $h_1, h_2 \in \mathcal{H}$ using unlabeled data, i.e., the fraction of examples on which the two checkpoints predict different labels. These estimates instantiate the constraints in Section 4.2, yielding the LP-based contribution vector $\mathbf{m}^{\mathrm{LP}}$.

**Experimental setup.** We assume $c_i = 1$ for all agents. Given $\mathbf{m}^{\mathrm{LP}}$, we perform an independent discrete search over integer allocations and, by repeated evaluation of the checkpoints in $\mathcal{H}$, find the smallest $\mathbf{m}^{\star}$ that satisfies every agent's $(\varepsilon, \delta)$ target. We sweep the size $|\mathcal{H}|$ of the sampled class and record $r$.

**Results.** The observed values of $r$ across $|\mathcal{H}|$ are reported in Table 2. The LP allocation achieves logarithmic-type approximation factors consistent with Theorem 3.

## B BINARY PAYOFFS AND QUASI-LINEAR UTILITIES

In this section we explain why our binary "meets target or not" payoff should be viewed as a con-venient normalization rather than a restrictive assumption. We model each agent's payoff as 1 if its PAC requirement holds and 0 otherwise, but show that this threshold formulation is equivalent, via a standard Lagrangian/KKT argument, to optimizing a much broader class of quasi-linear utilities that trade off accuracy against cost. In particular, any agent with smooth, monotone accuracy ben-efits and convex costs behaves as if it had such a hard accuracy requirement, so the binary model compactly represents a wide family of underlying preferences.

Table 2: LP total versus the optimal total on FEMNIST for a finite hypothesis class $\mathcal{H}$.

| $\|\mathcal{H}\|$ | $r = \frac{\sum_i m_i^{\mathrm{LP}}}{\sum_i m_i^\star}$ |
|---|---|
| 5 | 1.67 |
| 10 | 3.19 |
| 15 | 3.94 |
| 20 | 3.57 |
| 25 | 4.24 |
| 27 | 3.99 |
| 50 | 4.20 |

Our agent-level decision problem is a cost–minimization subject to meeting its accuracy requirement:

$$\min_{m_i \in \mathbb{N}} m_i \quad \text{s.t.} \quad a_i^{\varepsilon,\delta}(\boldsymbol{m}) = 1.$$

Under standard regularity conditions (feasibility of the constraint, monotonicity and differentiability of the underlying accuracy function, and convexity of the cost), the Karush–Kuhn–Tucker (KKT) conditions imply that any optimal solution to this constrained problem can be obtained by optimizing an equivalent unconstrained objective. Let $A_i(m_i)$ denote a smooth, increasing performance metric for agent $i$ (for example, one minus error) such that $A_i(m_i^\star) = 1$ at the optimum, and let the per-sample cost be $c_i > 0$. The Lagrangian for agent $i$'s problem is

$$\mathcal{L}_i(m_i, \lambda_i) = c_i m_i + \lambda_i\big(1 - A_i(m_i)\big),$$

with multiplier $\lambda_i \geq 0$ on the accuracy constraint. At any optimal solution $m_i^\star$, the KKT stationarity condition gives

$$\frac{\partial \mathcal{L}_i}{\partial m_i}(m_i^\star) = c_i - \lambda_i A_i'(m_i^\star) = 0 \quad \Rightarrow \quad A_i'(m_i^\star) = \frac{c_i}{\lambda_i}.$$

In particular, there exists a multiplier $\lambda_i > 0$ such that at the optimum $m_i^\star$, the marginal benefit of contributing data, $\mathcal{A}_i'(m_i^\star)$, equals the marginal cost $c_i$ scaled by $\lambda_i$. Intuitively, $\lambda_i$ serves to balance accuracy and cost at the threshold point. It is exactly the shadow price of the accuracy constraint: it measures how much the agent's minimum cost would increase if the required accuracy threshold were raised slightly.

This yields a precise equivalence between the threshold and quasi-linear viewpoints. Consider the unconstrained quasi-linear utility

$$U_i(m_i) = A_i(m_i) = \widetilde{\lambda}_i m_i,$$

where $\widetilde{\lambda}_i$ is the agent's marginal disutility for each contributed sample (we can absorb $c_i$ into $\widetilde{\lambda}_i$ by scaling). The first-order condition for maximizing $U_i$ is $A_i'(m_i) = \widetilde{\lambda}_i$. Choosing $\widetilde{\lambda}_i = c_i/\lambda_i$ makes this condition identical to the KKT stationarity condition above, so $m_i^\star$ is also optimal for the unconstrained problem. In other words, the same $m_i^\star$ solves both "minimize cost subject to meeting the threshold" and "maximize smooth quasi-linear utility that trades off accuracy and cost at rate $\widetilde{\lambda}_i$." The optimal stopping point for a satisficing agent under a continuous accuracy–benefit curve is therefore identical to the stopping point under the binary threshold model. Ex post, using a threshold utility *does not* exclude the optimal outcome that a more complex utility would produce: for any smooth, monotone accuracy–benefit function and convex cost, there exists a cost weight $\widetilde{\lambda}_i$ such that the agent behaves as if it had a hard requirement.

The above argument relies on $A_i(m_i)$ being monotone (more data never hurts) and differentiable; if we further assume $A_i$ is concave (diminishing returns on accuracy), the correspondence between the threshold model and a smooth utility model becomes tight. Under concavity, the stopping point $m_i^\star$ is uniquely characterized by the balance of marginal benefit and cost, and this point is exactly where it is optimal to "just meet" the target and not contribute further. In that sense, any agent with a smooth, monotone benefit of accuracy and convex cost behaves as if it had a personal threshold, and the binary payoff is a modeling convenience rather than a fundamental restriction.

## C    EXISTENCE AND EFFICIENCY OF EQUILIBRIA

In this section, we study the strategic behavior of self-interested agents in our model.

## C.1 EXISTENCE OF EQUILIBRIA

We impose weak *monotonicity*—more data from any agent never hurts another. The assumption guarantees equilibrium existence.

**Assumption 2** (Monotonicity). *For every agent $i$, utility is weakly increasing in any agent's contribution. Let $\mathbf{m}$ and $\mathbf{m}'$ be two contribution profiles with $m_j' \geq m_j$ for every agent $j$. Then*

$$u_i(\mathbf{m}') \ \geq \ u_i(\mathbf{m}) \quad \forall\, i \in \mathcal{A}..$$

**Theorem 5** (Existence of Nash Equilibrium). *A Nash equilibrium exists.*

*Proof.* Since every agent $i \in \mathcal{A}$ is self-sufficient, $i$ can, by contributing $n_i^{\mathrm{ind}} < \infty$ samples on their own, satisfy their objective. Since additional samples supplied by other agents never reduce any agent's probability of meeting its objective, the strategy $m_i = n_i^{\mathrm{ind}}$ guarantees utility $1 - c_i n_i^{\mathrm{ind}} \geq 0$ for every profile $\mathbf{m}_{-i}$. In contrast, any action with $c_i m_i > 1$ yields utility $1 - c_i m_i < 0$ irrespective of the others' choices, so every such action is strictly dominated by $n_i^{\mathrm{ind}}$. Thus agent $i$'s undominated actions lie in the finite set $\{0, 1, \ldots, n_i^{\mathrm{ind}}\}$. With each player restricted to finitely many pure strategies, the game is finite, and Nash's existence theorem for finite games guarantees at least one (possibly mixed) Nash equilibrium. $\quad\square$

We demonstrate, however, that pure Nash equilibria do not always exist:

**Theorem 6** (Non-existence of Pure Nash Equilibrium). *There exists a PAC learning setting in which no pure Nash equilibrium exists.*

*Proof.* Consider an instance space $\mathcal{X} = \{x_1, x_2, x_3\}$ and a hypothesis class $\mathcal{H}$ that contains all possible labeling. Let agents have data distributions: Define the data distributions of the three agents by the point-probabilities

$$\mathcal{D}_1(x_1) = \tfrac{1}{3},\ \mathcal{D}_1(x_2) = \tfrac{2}{3},\ \mathcal{D}_1(x_3) = 0,$$
$$\mathcal{D}_2(x_1) = 0,\ \mathcal{D}_2(x_2) = \tfrac{1}{3},\ \mathcal{D}_2(x_3) = \tfrac{2}{3},$$
$$\mathcal{D}_3(x_1) = \tfrac{2}{3},\ \mathcal{D}_3(x_2) = 0,\ \mathcal{D}_3(x_3) = \tfrac{1}{3}.$$

Fix the PAC parameters $\varepsilon = \tfrac{1}{3}$ and $\delta = \tfrac{2}{3}$. Each agent seeks accuracy at least $\tfrac{2}{3}$ with probability at least $\tfrac{1}{3}$. Hence every agent $i$ requires that, with probability at least $1 - \delta = \tfrac{1}{3}$, the learned classifier incurs error at most $\varepsilon$ on $\mathcal{D}_i$.

In this setup, no agent is incentivized to contribute more than one sample. Checking possible pure strategies, one observes: At $\mathbf{m} = (1, 1, 0)$, agent 1 can deviate to 0 without losing accuracy, incentivizing deviation. And, at $\mathbf{m} = (0, 1, 0)$, the contribution level is insufficient for agent 3. Thus, no pure equilibrium exists. $\quad\square$

## C.2 EFFICIENCY OF EQUILIBRIA: PRICE OF STABILITY

**Theorem 1** (PoS is unbounded). *For any $\varepsilon \in (0, 1)$ there exists a sequence of instances of our problem in which the ratio of the best NE to the optimal solution approaches $\Omega\big(\log(1/\varepsilon)\big)$.*

*Proof.* Fix $\varepsilon \in (0, \tfrac{1}{2})$ and $\delta \in (0, 1)$. We construct an instance whose PoS satisfies

$$\mathrm{PoS} \ = \ \frac{\log(1/\varepsilon) + \log(1/\delta)}{\log(1/\delta)} \ = \ \Omega\big(\log(1/\varepsilon)\big),$$

so the ratio diverges as $\varepsilon \to 0$. Let $n > \tfrac{1}{2\varepsilon}$ and set

$$\mathcal{X} \ = \ \{x_1, \ldots, x_n, y, z\}.$$

Define the hypothesis class

$$\mathcal{H} \ = \ \{h_0\} \ \cup \ \{h_i : i \in [n]\},$$

where

$$h_0(x) = 0 \quad (\forall\, x \in \mathcal{X}), \qquad h_i(x) = \begin{cases} 1 & \text{if } x \in \{x_i, z\}, \\ 0 & \text{otherwise.} \end{cases}$$

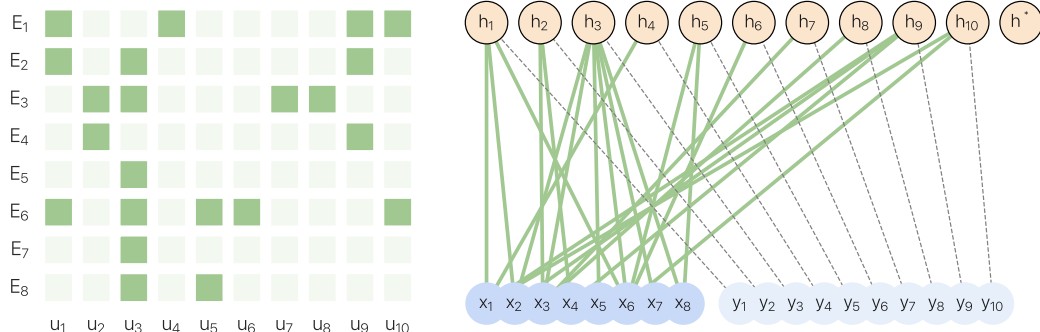

Figure 4: Visualization of the SET COVER reduction used in Theorem 2. **Left:** Incidence matrix of the original Set Cover instance. Each row is a subset $E_j$ and each column an element $u_i$; a square means $u_i \in E_j$. **Right:** The corresponding bipartite graph construction: each node $x_j$ corresponds to subset $E_j$ and is connected to the hypotheses $h_i$ for which $u_i \in E_j$. A size-$m$ set cover on the left corresponds to an $m$-sample training set that forces ERM to output $h^\star$ on the right.

Thus $y$ is always labeled $0$, while the label of $z$ determines whether all $x_1, \ldots, x_n$ are $0$ ($h^0$) or exactly one of them is $1$ ($h^i$).

Alice and Bob have marginal distributions

$$\mathcal{D}_{\text{Alice}}(x_i) = 2\varepsilon \quad \forall i \in [n], \qquad\qquad \mathcal{D}_{\text{Alice}}(y) = \mathcal{D}_{\text{Alice}}(z) = 0,$$

$$\mathcal{D}_{\text{Bob}}(z) = \varepsilon, \quad \mathcal{D}_{\text{Bob}}(y) = 1 - \varepsilon, \qquad \mathcal{D}_{\text{Bob}}(x_i) = 0 \quad \forall i \in [n].$$

In every NE, Bob contributes $0$ samples, because any $h \in \mathcal{H}$ has error at most $\varepsilon$ on his distribution, so his $(\varepsilon, \delta)$-requirement is already satisfied. In the worst case, where $h_0$ is the target function, Alice must choose $m$ so that she samples every point $x_1, \ldots, x_n$ with probability at least $1 - \delta$. Let $m_{\text{NE}}$ be the smallest integer for which this condition holds.

In the minimum-cost contribution vector, Bob supplies $m_{\text{Bob}}^{\text{opt}} = \frac{\log(1/\delta)}{\log(1-\varepsilon)}$ samples, which include $z$ with probability $1 - \delta$. If the true target is $h_0$ no further data are needed. Otherwise the target is some $h_i$; Alice then has to sample only until she sees the single point $x_i$. Let $m_{\text{Alice}}^{\text{OPT}}$ be the least integer guaranteeing this with probability $1 - \delta$. Sampling until one designated point appears is far cheaper than sampling until all $n$ points appear. Hence the optimal cost is

$$\|\mathbf{m}^{\text{opt}}\|_1 = m_{\text{Bob}}^{\text{opt}} + m_{\text{Alice}}^{\text{opt}} \leq \frac{3\log(1/\delta)}{2\varepsilon}.$$

Taking the ratio,

$$\text{PoS} = \frac{\|\mathbf{m}^{\text{NE}}\|_1}{\|\mathbf{m}^{\text{OPT}}\|_1} = \frac{\log(1/\varepsilon) + \log(1/\delta)}{\log(1/\delta)} = \Omega\big(\log(1/\varepsilon)\big),$$

which grows without bound as $\varepsilon \to 0$. $\qquad\square$

# D   DETAILS AND PROOFS FOR SECTION 4

## D.1   PROOF OF THE NP-HARDNESS RESULT (THEOREM 2)

We now formally address the computational complexity of deciding whether a given number of samples is sufficient to satisfy the $(\varepsilon, \delta)$-PAC learning guarantee for all agents. Given a data domain $\mathcal{X}$, a hypothesis class $\mathcal{H}$ of size $n$, distributions $\mathcal{D} = (\mathcal{D}_1, \ldots, \mathcal{D}_k)$ for each agent, an accuracy parameter $\varepsilon > 0$, a confidence parameter $\delta \in [0, 1)$, and an integer $m$, we define the *Collaborative PAC Sample-Allocation Problem* as follows:

**Definition 3** (Collaborative PAC Sample-Allocation Problem). *Given domain $\mathcal{X}$, hypothesis class $\mathcal{H}$, distributions $\mathcal{D} = (\mathcal{D}_1, \ldots, \mathcal{D}_k)$, accuracy parameter $\varepsilon > 0$, confidence parameter $\delta \in [0, 1)$, and integer $m$, decide whether drawing $m$ samples i.i.d. from each $\mathcal{D}_i$ suffices to guarantee, with probability at least $1 - \delta$, an ERM hypothesis of error at most $\varepsilon$ for every agent $i \in \mathcal{A}$.*

In other words, if each agent contributes exactly $m_i$ samples from their distribution, will the pooled dataset ensure $\varepsilon$-accurate performance on all distributions simultaneously? We now examine the computational complexity of this question. We establish the following computational hardness result:

**Theorem 2.** *Under the PAC-accuracy objective, determining whether a specified sample count $m$ suffices to meet the $(\varepsilon, \delta)$-requirement is NP-hard with respect to the hypothesis-class size $|\mathcal{H}|$, even when there is only one agent.*

*Proof.* We reduce from the NP-complete SET COVER problem (Garey & Johnson, 2002): given a universe $U = \{u_1, \ldots, u_n\}$, a family of subsets $\mathcal{E} = \{E_1, \ldots, E_r\}$ with $\bigcup_{i \in [r]} E_i = U$, and an integer $m$, decide whether at most $m$ subsets cover $U$.

We map an instance $(U, \mathcal{E}, m)$ of SET COVER to an instance of the Collaborative PAC Sample-Allocation Problem as follows: let the data domain be $\mathcal{X} = \mathcal{X}_1 \cup \mathcal{X}_2$ where

$$\mathcal{X}_1 = \{x_1, \ldots, x_r\}, \qquad \mathcal{X}_2 = \{y_1, \ldots, y_n\}.$$

Each point $x_j \in \mathcal{X}_1$ corresponds directly to subset $E_j \in \mathcal{E}$, and each point $y_i \in \mathcal{X}_2$ corresponds to element $u_i \in U$. Define the hypothesis class

$$\mathcal{H} = \{h^\star, h_1, \ldots, h_n\},$$

where, for each $i \in [n]$ and $j \in [k]$:

$$h_i(x_j) = \begin{cases} 1 & \text{if } u_i \in E_j \\ 0 & \text{otherwise} \end{cases}, \quad h_i(y_j) = \begin{cases} 1 & \text{if } i = j \\ 0 & \text{otherwise} \end{cases}, \quad h^\star(x_j) = 0, \quad h^\star(y_j) = 0.$$

Set the distribution $\mathcal{D}$ to be uniform over $\mathcal{X}$. Choose $0 < \varepsilon < 1/(r+n)$ so that a single classification error violates the $\varepsilon$-accuracy goal, and set $\delta$ with $1 - \delta \geq (|\mathcal{X}|^{-1})^m$.

If a set cover of size $p \leq m$ exists, say $\{E_{j_1}, \ldots, E_{j_p}\}$, consider the sample $S = \{x_{j_1}, \ldots, x_{j_p}\}$. For every competing hypothesis $h_i \neq h^\star$ there is a subset $E_{j_q}$ containing $u_i$, hence $h_i(x_{j_q}) = 1 \neq 0 = h^\star(x_{j_q})$. Each $h_i$ therefore incurs at least one error on $S$, so ERM selects $h^\star$. Thus $m$ samples suffice to meet the agent's $(\varepsilon, \delta)$-requirement.

Conversely, assume that $m$ samples drawn i.i.d. from $\mathcal{D}$ suffice to guarantee, with probability at least $1 - \delta$, an ERM hypothesis achieving error at most $\varepsilon$ for any possible true hypothesis. Due to our choice of $\varepsilon$, the ERM hypothesis must correctly classify all points in $\mathcal{X}$. Consider the worst-case scenario where the true hypothesis is $h^\star$. To distinguish $h^\star$ from all other hypotheses $h_i$, the training set must include points from $\mathcal{X}_1$ that differentiate $h^\star$ from each $h_i$. Specifically, for each element $u_i \in U$, the training set must contain at least one point $x_j \in \mathcal{X}_1$ with $h_i(x_j) = 1$ (meaning $u_i \in E_j$), ensuring $h_i$ is eliminated by ERM. Therefore, the set of points in the minimal training set corresponds directly to subsets forming a valid set cover of size at most $m$.

Notably, if any other hypothesis $h_i \neq h^\star$ were the true target, fewer samples would trivially suffice. Thus, the case where $h^\star$ is the true hypothesis indeed represents the worst-case. $\qquad\square$

### D.2 PROOF OF THE APPROXIMATION GUARANTEE (THEOREM 3)

**Theorem 3** (Approximation via Linear Programming). *Given any $\mathcal{H}$ and $\varepsilon, \delta > 0$:*

- *For finite $\mathcal{H}$, the LP over $(\mathcal{H}, \varepsilon, \delta)$ returns a $\frac{\log(1/\delta) + \log|\mathcal{H}|}{\log(1/\delta)}$-approximate solution to Equation 4.*

- *For infinite $\mathcal{H}$, running the LP over $(\overline{\mathcal{H}}, \varepsilon, \delta')$ and multiplying it by $d + \log(1/\delta'')$ returns a $O(\frac{d^2 (\log(c_{max}kd/(c_{min}\varepsilon\delta))^2}{\log(1/\delta)})$-approximate solution to Equation 4, where $\overline{\mathcal{H}} \subset \mathcal{H}$ is a $\gamma$-cover of $\mathcal{H}$, $\gamma = \Theta(c_{min}\varepsilon\delta/(c_{max}k(d + \log(1/\delta))))$, $d = \mathrm{VCdim}(\mathcal{H})$, $\delta'' = \frac{\delta}{4|\overline{\mathcal{H}}|}$ and $\delta' = \frac{\delta}{8(d + \log(2|\overline{\mathcal{H}}|/\delta))}$, $c_{min} = \min_{i \in [k]} c_i$ and $c_{max} = \max_{i \in [k]} c_i$.*

*Proof.* We start with the case where the hypothesis class $\mathcal{H}$ is finite, the extension to infinite $\mathcal{H}$ follows with a standard covering argument.

**Proof for finite** $\mathcal{H}$**.** For any pair of hypotheses $(h_1, h_2) \in \mathcal{H}$, define the *disagreement regions* between $h_1$ and $h_2$ as

$$\mathrm{DIS}(h_1, h_2) \;=\; \{\, x \in \mathcal{X} \;\mid\; h_1(x) \neq h_2(x) \,\}.$$

This region represents the set of points where $h_1$ and $h_2$ disagree. For each agent $i$, let $\mathcal{D}_i\big(\mathrm{DIS}(h_1, h_2)\big)$ be the probability mass that $\mathcal{D}_i$ places on that region.

We want to guarantee that with probability at least $1 - \delta$, the ERM solution has error $\leq \varepsilon$ for all agents simultaneously. Concretely, if the true labeler is $h^\star \in \mathcal{H}$, then *any* other hypothesis $h_2$ that has $\mathcal{D}_i(\mathrm{DIS}(h^\star, h_2)) > \varepsilon$ (meaning it is "bad" for agent $i$) must be *eliminated* by at least one sample from that disagreement region.

The probability that $h_2$ is never eliminated, given $h^\star$ is the true labeler, is exactly the probability that *no sample* from any agent $i$ ever lands in $\mathrm{DIS}(h^\star, h_2)$. Since each agent $i$ contributes $m_i$ i.i.d. points from $\mathcal{D}_i$,

$$\Pr\big(h_2 \text{ is not eliminated} \;\mid\; h^\star = h^\star\big) \;=\; \prod_{i=1}^{k}\Big(1 - \mathcal{D}_i\big(\mathrm{DIS}(h^\star, h_2)\big)\Big)^{m_i}.$$

Hence the requirements $\Pr(h_2 \text{ not eliminated} \mid h^\star) \leq \frac{\delta}{H}$ are log-linear constraints:

$$\sum_{i=1}^{k} m_i \, \log\!\Big[\, 1 - \mathcal{D}_i\big(\mathrm{DIS}(h^\star, h_2)\big)\Big] \;\leq\; \log\!\Big(\tfrac{\delta}{H}\Big).$$

We impose such constraints for *all* pairs $(h^\star, h_2)$ where $\mathcal{D}_i(\mathrm{DIS}(h^\star, h_2)) > \varepsilon$ for agent $i$. By union bounding across $\leq H$ possible "bad" hypotheses $h_2$ (for each $h^\star$) or $\leq H^2$ pairs overall, we ensure that with probability $\geq 1 - \delta$, any truly bad hypothesis (in the sense of having error $> \varepsilon$) is eliminated.

Hence, we obtain a linear constraint in terms of $m_i$, which allows us to approximate the elimination probability for each hypothesis pair $(h_1, h_2)$ by solving the following linear program:

$$\min_{\mathbf{m} \in \mathbb{N}^k} \quad \mathbf{c}^\top \mathbf{m}$$

subject to

$$\sum_{i=1}^{k} m_i \log\left(1 - \mathcal{D}_i(\mathrm{DIS}(h_1, h_2))\right) \leq \log \frac{\delta}{H} \quad \forall (h_1, h_2) \in \mathcal{H}^2 \text{ s.t. } \exists i \in [k], \mathcal{D}_i(\mathrm{DIS}(h_1, h_2)) > \varepsilon.$$

$$(7)$$

One can then round the fractional solution up to integer counts $m_i$.

Let $\mathbf{m}^*$ be the solution of the linear program. For any agent $i$, we want to show:

$$\forall h^\star \in \mathcal{H}, \qquad \Pr_{U_j \sim \mathcal{D}_j^{m_j^*}, j \in [k]} \left(\mathrm{err}_{\mathcal{D}_i}^{\mathrm{ERM}}\left(\bigcup_{j \in [k]} U_j \times h^\star(U_j)\right) > \varepsilon\right) \leq \delta.$$

This holds because the linear program enforces that, for each pair $(h_1, h_2) \in \mathcal{H}^2$, the probability of failing to eliminate any incorrect hypothesis $h_2$ (given $h^\star = h_1$) is bounded by $\delta/H$. By applying a union bound over all hypothesis pairs, we achieve the desired bound.

To show that this solution is $\frac{\log(1/\delta) + \log H}{\log(1/\delta)}$-approximate optimal, let $\mathbf{m}$ be the true optimal solution. For each pair $(h_1, h_2)$ such that $\mathcal{D}_i(\mathrm{DIS}(h_1, h_2)) > \varepsilon$, if we take $\mathbf{m}$ samples, we have:

$$\prod_{i=1}^{k} (1 - \mathcal{D}_i(\mathrm{DIS}(h_1, h_2)))^{m_i} \leq \delta.$$

Taking logarithms we obtain:

$$\sum_{i=1}^{k} m_i \log\left(1 - \mathcal{D}_i(\mathrm{DIS}(h_1, h_2))\right) \leq \log \delta.$$

Now, given a sample size of $\frac{\log(1/\delta)+\log H}{\log(1/\delta)} \cdot \mathbf{m}$, we have:

$$\sum_{i=1}^{k} \frac{\log(1/\delta) + \log H}{\log(1/\delta)} \cdot m_i \log\left(1 - \mathcal{D}_i(\mathrm{DIS}(h_1, h_2))\right) \leq \log\frac{\delta}{H},$$

which satisfies the constraint in Equation 7.

Next, we show how to lift the guarantee to infinite classes using $\gamma$-covers and apply our results to this finite cover.

**Proof for infinite $\mathcal{H}$.** For any $\gamma \in (0,1)$, we say $\overline{\mathcal{H}}$ is a $\gamma$-cover of $\mathcal{H}$ under $\boldsymbol{\mathcal{D}}$ if for all $h \in \mathcal{H}$, there exists an $\overline{h} \in \overline{\mathcal{H}}$ satisfying that $\max_i \mathcal{D}_i(\mathrm{DIS}(h, \overline{h})) \leq \gamma$.

**Lemma 2.** *Let $d = \mathrm{VCdim}(\mathcal{H})$ denote the VC dimension of $\mathcal{H}$. There is a $\gamma$-cover $\overline{\mathcal{H}} \subset \mathcal{H}$ under $\boldsymbol{\mathcal{D}}$ of size $\left(\frac{41k}{\gamma}\right)^d$.*

*Proof of Lemma 2.* In learning theory, for any distribution $\mathcal{D}$, a subset $\overline{\mathcal{H}} \subseteq \mathcal{H}$ is a $\gamma$-cover of $\mathcal{H}$ under $\mathcal{D}$ if, for every $h \in \mathcal{H}$, there exists $\overline{h} \in \overline{\mathcal{H}}$ such that $\mathcal{D}(\mathrm{DIS}(h, \overline{h})) \leq \gamma$. Haussler's sphere-packing bound guarantees Haussler (1995) the existence of such a cover with

$$|\overline{\mathcal{H}}| \leq \left(\frac{41}{\gamma}\right)^d.$$

Construct a $\frac{\gamma}{k}$-cover of $\mathcal{H}$ under the averaged distribution $\frac{1}{k}\sum_{i=1}^{k} \mathcal{D}_i$. For any $h \in \mathcal{H}$ and its representative $\overline{h}$ in this cover,

$$\max_i \mathcal{D}_i\left(\mathrm{DIS}(\overline{h}, h)\right) \leq \sum_{i=1}^{k} \mathcal{D}_i\left(\mathrm{DIS}(\overline{h}, h)\right) \leq \gamma,$$

so the same set is a $\gamma$-cover under $\boldsymbol{\mathcal{D}} = (\mathcal{D}_1, \ldots, \mathcal{D}_k)$. Substituting $\frac{\gamma}{k}$ into Haussler's bound yields

$$|\overline{\mathcal{H}}| \leq \left(\frac{41k}{\gamma}\right)^d.$$

$\square$

Let $\boldsymbol{m}^{\overline{\mathcal{H}}, \delta'}$ denote the solution to the LP (Equation 7) given hypothesis class $\overline{\mathcal{H}}$ and confidence parameter $\delta'$. Let $c_{\min} = \min_{i \in [k]} c_i$ and $c_{\max} = \max_{i \in [k]} c_i$.

**Lemma 3.** *By choosing $\gamma = \Theta\left(c_{min}\varepsilon\delta/(c_{max}k(d + \log(1/\delta)))\right)$ and $\boldsymbol{m} = \boldsymbol{m}^{\overline{\mathcal{H}}, \delta/2}$, for any target hypothesis $h^\star \in \mathcal{H}$ and agent $i$, with probability at least $1 - \delta$, any consistent hypothesis $\overline{h} \in \overline{\mathcal{H}}$ will satisfy*

$$\mathrm{err}_{\mathcal{D}_i, h^\star}(\overline{h}) \leq \varepsilon.$$

*Proof of Lemma 3.* Since each agent can achieve the PAC learning objective with at most $O((d + \log(1/\delta))/\varepsilon)$ data points individually, we restrict attention to contribution vectors satisfying $\|\mathbf{m}\|_1 \leq O(\frac{c_{\max}k(d+\log(1/\delta))}{c_{\min}\varepsilon})$ (otherwise we can replace the solution with $O((d + \log(1/\delta))/\varepsilon)\mathbf{1}$). For any $\mathbf{m}$ satisfying this constraint, any $h^\star \in \mathcal{H}$, and any agent $i$, let the labeled sample be

$$S^{h^\star} = \bigcup_{j \in [k]} \{(x, h^\star(x)) \mid x \in U_j\}.$$

Define

$$\overline{\mathrm{err}} = \max\left\{\mathrm{err}_{\mathcal{D}_i}(\overline{h}) \,\middle|\, \overline{h} \in \overline{\mathcal{H}}, \ \mathrm{err}_{S^{h^\star}}(\overline{h}) = \min_{h' \in \overline{\mathcal{H}}} \mathrm{err}_{S^{h^\star}}(h')\right\},$$

the worst-case error when running ERM over $\overline{\mathcal{H}}$. We can show that $\overline{\mathrm{err}}$ is small with high probability. More specifically, assuming a $(\varepsilon, \delta/2)$-PAC guarantee for $\overline{\mathcal{H}}$, we have

$$
\Pr_{S^{h^\star}}\Big[\exists \overline{h} : \mathrm{err}_{\mathcal{D}_i}(\overline{h}) > \varepsilon, \ \mathrm{err}_{S^{h^\star}}(\overline{h}) = \min_{h' \in \overline{\mathcal{H}}} \mathrm{err}_{S^{h^\star}}(h')\Big]
$$

$$
\leq \Pr_{S^{h^\star}}\Big[\exists \overline{h} : \mathrm{err}_{\mathcal{D}_i}(\overline{h}) > \varepsilon, \ \mathrm{err}_{S^{h^\star}}(\overline{h}) = 0, \ \mathrm{err}_{S^{h^\star}}(\overline{h}^\star) = 0\Big] + \Pr_{S^{h^\star}}\Big[\mathrm{err}_{S^{h^\star}}(\overline{h}^\star) \neq 0\Big]
$$

$$
\leq \Pr_{S^{h^\star}}\Big[\exists \overline{h} : \mathrm{err}_{\mathcal{D}_i}(\overline{h}) > \varepsilon, \ \mathrm{err}_{S^{h^\star}}(\overline{h}) = 0\Big] + \Pr_{S^{h^\star}}\Big[\mathrm{err}_{S^{h^\star}}(\overline{h}^\star) \neq 0\Big]
$$

$$
\leq \frac{\delta}{2} + \big(1 - (1-\gamma)^{\sum_j m_j}\big) \ \leq \ \frac{\delta}{2} + \big(1 - (1-\gamma)^{\frac{c_{\max}k(d+\log(1/\delta))}{c_{\min}\varepsilon}}\big).
$$

Choosing $\gamma = c_{\min}\varepsilon\delta/(c_{\max}k(d + \log(1/\delta)))$ yields

$$
\frac{\delta}{2} + \Big(1 - (1-\gamma)^{\frac{c_{\max}k(d+\log(1/\delta))}{c_{\min}\varepsilon}}\Big) \ \leq \ \delta,
$$

so for any $h^\star$ and agent $i$, with probability at least $1 - \delta$ we can always find a hypothesis whose error is at most $\varepsilon$. $\qquad\square$

**Lemma 4.** *By choosing $\gamma = \Theta\big(c_{min}\varepsilon\delta/(c_{max}k(d+\log(1/\delta)))\big)$, the solution $(d+\log(1/\delta'')) \cdot \mathbf{m}^{\overline{\mathcal{H}}, \delta'}$ is sufficient is sufficient to achieve $(\varepsilon, \delta)$-PAC accuracy objective ([Equation 2](#)) for $\mathcal{H}$, where $\delta'' = \frac{\delta}{4|\overline{\mathcal{H}}|}$ and $\delta' = \frac{\delta}{8(d+\log(2|\overline{\mathcal{H}}|/\delta))}$.*

*Proof of [Lemma 4](#).* Similar to [Lemma 3](#), we again restrict attention to contribution vectors satisfying $\|\mathbf{m}\|_1 \leq O\big(\frac{c_{\max}k(d+\log(1/\delta))}{c_{\min}\varepsilon}\big)$. This guarantee that for any target $h^\star$, with probability at least $1 - \frac{\delta}{2}$, $\overline{h}^\star$ is consistent. Hence, we can view $\overline{h}^\star$ as our target hypothesis.

We now prove that for any target hypothesis $\overline{h}^\star \in \overline{\mathcal{H}}$ and any representative hypothesis $\overline{h} \in \overline{\mathcal{H}}$ satisfying $\exists i \in [k], \mathcal{D}_i(\mathrm{DIS}(\overline{h}^\star, \overline{h})) > \varepsilon$, with probability at least $1 - \frac{\delta'}{2|\overline{\mathcal{H}}|}$, any $h$ in the $\gamma$-ball of $\overline{h}$ will be eliminated.

Let $\beta_i = \mathcal{D}_i(\mathrm{DIS}(\overline{h}^\star, \overline{h}))$ denote the probability mass of $\mathrm{DIS}(\overline{h}^\star, \overline{h})$ under agent $i$'s data distribution and let $A_\gamma = \{i \in [k] | \beta_i \geq 4\gamma\}$ denote the set of agents whose probability mass of the disagreement region is at least $4\gamma$. The approximation solution $\boldsymbol{m}^{\overline{\mathcal{H}}, \delta'}$ guarantees that

$$
\prod_{i=1}^{k} (1 - \beta_i)^{m_i^{\overline{\mathcal{H}}, \delta'}} \leq \frac{\delta'}{|\overline{\mathcal{H}}|}.
$$

If $\beta_i < 4\gamma$, we have

$$
(1 - \beta_i)^{m_i^{\overline{\mathcal{H}}, \delta'}} > (1 - 4\gamma)^{m_i^{\overline{\mathcal{H}}, \delta'}} \geq (1 - 4\gamma)^{\delta'/\gamma} \geq e^{-(4\log 4)\delta'},
$$

where we adopt the inequality $(1 - 1/x)^x \geq \frac{1}{4}$ for all $x \geq 2$. Hence, we have

$$
e^{-(4\log 4)k\delta'} \cdot \prod_{i:\beta_i \geq 4\gamma} (1 - \beta_i)^{m_i^{\overline{\mathcal{H}}, \delta'}} \leq \frac{\delta'}{|\overline{\mathcal{H}}|}.
$$

By rearranging terms, we have

$$
\prod_{i:\beta_i \geq 4\gamma} (1 - \beta_i)^{m_i^{\overline{\mathcal{H}}, \delta'}} \leq \frac{\delta'}{|\overline{\mathcal{H}}|} \cdot e^{(4\log 4)k\delta'} \leq \frac{2\delta'}{|\overline{\mathcal{H}}|},
$$

when $\delta' = O(\frac{1}{k})$. That is to say, even if we only collect $m_i$ samples from agent $i \in A_\gamma$ and don't collect from $i \notin A_\gamma$, we will still be able to obtain one sample from the $\mathrm{DIS}(\overline{h}^\star, \overline{h})$ with probability at least $1 - \frac{2\delta'}{|\overline{\mathcal{H}}|}$.

For any distribution $\mathcal{D}$, let $\mathcal{D}_{|\overline{h}^\star, \overline{h}}$ denote the distribution restricted to $\mathrm{DIS}(\overline{h}^\star, \overline{h})$, i.e.,

$$
\mathcal{D}_{|\overline{h}^\star, \overline{h}}(x) = \begin{cases} \frac{\mathcal{D}(x)}{\mathcal{D}(\mathrm{DIS}(\overline{h}^\star, \overline{h}))} & \text{if } x \in \mathrm{DIS}(\overline{h}^\star, \overline{h}), \\ 0 & \text{otherwise.} \end{cases}
$$

Consider $\mathcal{D} = \frac{\sum_{i \in A_\gamma} m_i \mathcal{D}_i}{\sum_{i \in A_\gamma} m_i}$ being a mixture of $\{\mathcal{D}_i | i \in A_\gamma\}$, the disagreement region between $\overline{h}^\star$ and $h$ under distribution $\mathcal{D}_{|\overline{h}^\star, \overline{h}}$ is at least $\frac{3}{4}$. This is because

$$\mathcal{D}_{|\overline{h}^\star, \overline{h}}(\mathrm{DIS}(\overline{h}^\star, h)) = 1 - \mathcal{D}_{|\overline{h}^\star, \overline{h}}(\mathrm{DIS}(\overline{h}, h)) \geq 1 - \frac{\gamma}{4\gamma} = \frac{3}{4}.$$

Hence, by standard PAC learning guarantee, if we obtain $n = O(d + \log(1/\delta''))$ samples from $\mathcal{D}_{|\overline{h}^\star, \overline{h}}$, then with probability $1 - \delta''$, all $h$'s in the $\gamma$-ball of $\overline{h}$ are not consistent.

The current approximation solution $\mathbf{m}^{\overline{\mathcal{H}}, \delta'}$ can only guarantee that with probability $1 - \frac{2\delta'}{|\overline{\mathcal{H}}|}$, we obtain one sample from $\mathcal{D}_{|\overline{h}^\star, \overline{h}}$, i.e.,

$$\prod_{i: \beta_i \geq 4\gamma} (1 - \beta_i)^{m_i} \leq \frac{2\delta'}{|\overline{\mathcal{H}}|}.$$

When we increase $\mathbf{m}^{\overline{\mathcal{H}}, \delta'}$ by $n$ times, then with probability at least $1 - \frac{2n\delta'}{|\overline{\mathcal{H}}|}$, we obtain $n$ samples from $\mathcal{D}_{|\overline{h}^\star, \overline{h}}$.

Hence, by setting $\delta'' = \frac{\delta}{4|\overline{\mathcal{H}}|}$ and $\delta' = \frac{\delta}{8(d + \log(2|\overline{\mathcal{H}}|/\delta))}$, solving the LP for $\overline{\mathcal{H}}, \delta'$ to obtain an approximate solution $\mathbf{m}^{\overline{\mathcal{H}}, \delta'}$ and multiplying it by $d + \log(1/\delta'')$, the solution $(d + \log(1/\delta'')) \cdot \mathbf{m}^{\overline{\mathcal{H}}, \delta'}$ is sufficient to achieve $(\varepsilon, \delta)$-PAC accuracy objective ( Equation 2) for $\mathcal{H}$. $\qquad\square$

The solution $\mathbf{m}^{\overline{\mathcal{H}}, \delta'}$ returned by the approximation algorithm for the $(\varepsilon, \delta')$-PAC objective for $\overline{\mathcal{H}}$ is at most a factor

$$\frac{\log(|\overline{\mathcal{H}}|/\delta')}{\log(1/\delta)} = \frac{\log(8/\delta) + \log(d + \log(2|\overline{\mathcal{H}}|/\delta)) + d \log(k(d + \log(1/\delta))/(\varepsilon\delta))}{\log(1/\delta)}$$

$$= O\left(\frac{d(\log k + \log d + \log(1/\varepsilon) + \log(1/\delta) + \log(c_{\max}/c_{\min}))}{\log(1/\delta)}\right)$$

larger than the optimal solution for the $(\varepsilon, \delta)$-PAC objective for $\overline{\mathcal{H}}$. Since the PAC objective for $\overline{\mathcal{H}}$ is easier than the PAC objective for $\mathcal{H}$, we have $(d + \log(1/\delta'')) \cdot \mathbf{m}^{\overline{\mathcal{H}}, \delta'}$ is a $O\left(\frac{d^2(\log k + \log d + \log(1/\varepsilon) + \log(1/\delta) + \log(c_{\max}/c_{\min}))^2}{\log(1/\delta)}\right)$-approximation solution for $(\varepsilon, \delta)$-PAC objective for $\mathcal{H}$.

$\qquad\square$

# E  DETAILS AND PROOFS FOR SECTION 5

**Lemma 1.** *For any $H$ greater than a universal constant $C$, there exists a PAC learning instance of $(\mathcal{H}, \varepsilon, \delta)$ in the two-agent setting with $|\mathcal{H}| = H$ such that the approximation algorithm introduced in Theorem 3 is oblivious at all $\mathbf{m}$ with $m_1, m_2 \geq 2|\mathcal{H}| \log |\mathcal{H}|$.*

*Proof of Lemma 1.* Consider $\mathcal{X} = \{1, 2, \ldots, n\}$ as our domain and the hypothesis class $\mathcal{H}$ of *all singletons* (i.e. functions $h_i$ that label exactly one $x \in \mathcal{X}$ as positive and all others as negative) *plus* the all-negative function. Thus $H := |\mathcal{H}| = n + 1$.

We now show that that for any $\mathbf{m}$ satisfying $m_1, m_2 \geq 2|\mathcal{H}| \log |\mathcal{H}|$ and any neighbor $\mathbf{m}'$ with $\|\mathbf{m}' - \mathbf{m}\|_1 = 1$, we can always find distributions $\mathcal{D}_1, \mathcal{D}_1', \mathcal{D}_2, \mathcal{D}_2'$ satisfying that

- $\mathbf{m} = \mathrm{APPROX}(\mathcal{D}_1, \mathcal{D}_2)$ and $\mathbf{m}' = \mathrm{APPROX}(\mathcal{D}_1', \mathcal{D}_2) = \mathrm{APPROX}(\mathcal{D}_1, \mathcal{D}_2')$.
- Contributions $\mathbf{m}$ and $\mathrm{APPROX}(\mathcal{D}_1', \mathcal{D}_2)'$ are both feasible for $(\mathcal{D}_1, \mathcal{D}_2)$, $(\mathcal{D}_1', \mathcal{D}_2)$, and $(\mathcal{D}_1, \mathcal{D}_2')$.

**Construction.**  We construct a pair of distributions $(\mathcal{D}_1, \mathcal{D}_2)$ by selecting $p_1, p_2, q_1, q_2$ satisfying

$$p_1 \leq p_2 \leq \frac{1}{2n}, \quad q_2 \leq q_1 \leq \frac{1}{2n}, \quad p_1 + p_2 \leq \frac{1}{n}, \quad q_1 + q_2 \leq \frac{1}{n}.$$

Then let

$$p_3 = \ldots = p_n = \frac{1 - (p_1 + p_2)}{n - 2}, \quad q_3 = \ldots = q_n = \frac{1 - (q_1 + q_2)}{n - 2}.$$

Hence let $\mathcal{D}_1 = (p_1, \ldots, p_n)$ and $\mathcal{D}_2 = (q_1, \ldots, q_n)$. Both are well-defined distributions. We denote by $\mathcal{P}$ the family of all such pairs $(\mathcal{D}_1, \mathcal{D}_2)$ with different choices of $p_1, p_2, q_1, q_2$ satisfying the above constraints.

The approximate-optimal solution $\text{APPROX}(\mathcal{D}_1, \mathcal{D}_2)$ is found by solving a linear program derived from disagreement regions. For example, the disagreement region between the all-negative hypothesis $h_0$ and a singleton $h_1$ that is positive on point 1 has measure $p_1$ in $\mathcal{D}_1$ (and $q_1$ in $\mathcal{D}_2$). The "binding constraints" come from pairs $(h_0, h_1)$ and $(h_0, h_2)$ (the singletons with points 1 or 2). By contrast, any singleton on point $i \geq 3$ yields a disagreement measure at least $p_3$, which is larger, and thus any feasible solution that satisfies the "small measure" constraints with some margin also satisfies these larger measure constraints.

Thus the main LP constraints reduce to requiring that

$$m_1 \log \frac{1}{1 - p_1} + m_2 \log \frac{1}{1 - q_1} \geq \log\left(\frac{H}{\delta}\right), \tag{8}$$

$$m_1 \log \frac{1}{1 - p_2} + m_2 \log \frac{1}{1 - q_2} \geq \log\left(\frac{H}{\delta}\right). \tag{9}$$

The solution $\text{APPROX}(\mathcal{D}_1, \mathcal{D}_2)$ is the one that *minimizes* $c_1 \cdot m_1 + c_2 \cdot m_2$ subject to Equations (8) and (9) (and additional constraints for any bigger disagreements, satisfied by slack).

If $m_1 + m_2 \geq 2H \log H$, we can show:

$$m_1 \log\left(\frac{1}{1-p_i}\right) + m_2 \log\left(\frac{1}{1-q_i}\right) > m_1 \log\left(\frac{1}{1-p_1}\right) + m_2 \log\left(\frac{1}{1-q_1}\right) + \log H$$

for each $i \geq 3$, hence those constraints are looser. Hence, satisfying Equations (8) and (9) by a small margin also satisfies the bigger-disagreement constraints.

**To satisfy the second bullet.** For any $i = 3, 4, \ldots$, according to our construction, we have $p_i \geq \frac{1}{n}$ and $p_1, p_2 \leq \frac{1}{2n}$. Thus, we have

$$1 - p_1 \geq 1 - \frac{1}{2n} = e^{-\frac{1}{2n}} - O\left(\frac{1}{n^2}\right) = e^{-\frac{1}{n}} e^{\frac{1}{2n}} - O\left(\frac{1}{n^2}\right) \geq \left(1 - \frac{1}{n}\right) e^{\frac{1}{2n}} - O\left(\frac{1}{n^2}\right)$$

$$\geq (1 - p_i) e^{\frac{1}{2n}} - O\left(\frac{1}{n^2}\right).$$

Thus, we have

$$\log\left(\frac{1}{1 - p_i}\right) \geq \log\left(\frac{1}{1 - p_1}\right) + \frac{1}{2n} - O\left(\frac{1}{n^2}\right) \geq \log\left(\frac{1}{1 - p_1}\right) + \frac{1}{3n}$$

for $n$ big enough. Then for the constraint corresponding to the disagreement region $\text{DIS}(h_0, h_i)$, we have

$$m_1 \log \frac{1}{1 - p_i} + m_2 \log \frac{1}{1 - q_i} \geq m_1 \log \frac{1}{1 - p_1} + m_2 \log \frac{1}{1 - q_1} + \frac{m_1 + m_2}{3n}$$

$$\geq m_1 \log \frac{1}{1 - p_1} + m_2 \log \frac{1}{1 - q_1} + \log H,$$

where the last inequality holds when $m_1 + m_2 \geq 3H \log H$. Similar results also hold for $\mathcal{D}_2$. That is to say, instead of satisfying Equations (8) and (9), it would be sufficient to satisfy

$$m_1 \log \frac{1}{1 - p_1} + m_2 \log \frac{1}{1 - q_1} \geq \log \frac{3}{\delta},$$

$$m_1 \log \frac{1}{1 - p_2} + m_2 \log \frac{1}{1 - q_2} \geq \log \frac{3}{\delta}.$$

Hence, for any $(\mathcal{D}_1, \mathcal{D}_2) \in \mathcal{P}$, we have $\frac{\log 3/\delta}{\log H/\delta} \cdot \text{APPROX}(\mathcal{D}_1, \mathcal{D}_2)$ is feasible. For simplicity, let's fix $\delta \leq 0.5$ and suppose $H \geq 18$ from now. Then $\frac{1}{2} \cdot \text{APPROX}(\mathcal{D}_1, \mathcal{D}_2)$ is sufficient and so is $\text{APPROX}(\mathcal{D}_1, \mathcal{D}_2) - (1, 1)$. Thus, we justify bullet 2.

**To satisfy the first bullet.** Given an $\mathbf{m}$ and $\mathbf{m}'$, we pick $\mathcal{D}_1$ specified by $p_1, p_2$, $\mathcal{D}_2$ specified by $q_1, q_2$ and $\mathcal{D}_1'$ specified by $p_1', p_2'$ so that Inequalities Equations (8) and (9) hold with equality and that $\mathbf{m}$ and $\mathbf{m}'$ is the only solution to these linear equalities w.r.t. $(p_1, p_2, q_1, q_2)$ and w.r.t. $(p_1', p_2', q_1, q_2)$, respectively.

Inequalities Equations (8) and (9) can be approximated using a first-order approximation as follows.

$$m_1 p_1 + m_2 q_1 = \log \frac{H}{\delta} =: \alpha\,,$$

$$m_1 p_2 + m_2 q_2 = \alpha\,,$$

$$m_1' p_1' + m_2' q_1 = \alpha\,,$$

$$m_1' p_2' + m_2' q_2 = \alpha\,.$$

Let's pick

$$p_1 = \frac{\alpha}{m_1 m_2}, \qquad q_1 = \frac{(1 - 1/m_2)\alpha}{m_2}\,,$$

$$p_2 = \frac{(1 - 1/m_1)\alpha}{m_1}, \qquad q_2 = \frac{\alpha}{m_1 m_2}\,.$$

We can justify $(\mathcal{D}_1, \mathcal{D}_2) \in \mathcal{P}$ since $m_1, m_2 \geq 2n \log n$. Then if $\mathbf{m}'$ differs from $\mathbf{m}$ at $m_1$, by solving

$$m_1' p_1' + m_2 q_1 = m_1 p_1 + m_2 q_1\,,$$

$$m_1' p_2' + m_2 q_2 = m_1 p_2 + m_2 q_2\,,$$

we have

$$p_1' = \frac{m_1 p_1}{m_1'}, \qquad p_2' = \frac{m_1 p_2}{m_1'}\,.$$

It's easy for us to justify that $(\mathcal{D}_1', \mathcal{D}_2) \in \mathcal{P}$ since $\mathcal{D}_1'$ is very close to $\mathcal{D}_1$.

If $m'$ differs from $m$ at $m_2$, by solving

$$m_1 p_1' + m_2' q_1 = m_1 p_1 + m_2 q_1\,,$$

$$m_1 p_2' + m_2' q_2 = m_1 p_2 + m_2 q_2\,,$$

we have

$$p_1' = \frac{(m_2 - m_2')q_1}{m_1} + p_1, \qquad p_2' = \frac{(m_2 - m_2')q_2}{m_1} + p_2\,.$$

By plugging in the values of $p_1, p_2, q_1, q_2$, we have

$$p_1' = \frac{(\pm 1)(1 - 1/m_2)\alpha}{m_1 m_2} + \frac{\alpha}{m_1 m_2}\,,$$

$$p_2' = \frac{(\pm 1)\alpha}{m_1^2 m_2} + \frac{(1 - 1/m_1)\alpha}{m_1}\,.$$

It is easy to see that $p_2'$ is very close to $p_2$ and $p_1' < p_2'$. For $p_1'$, we need to make it fall in $(0, \frac{1}{2n})$. When $m_1, m_2 \geq 2n \log n$, we have $p_1' \leq \frac{2}{m_1 m_2} < \frac{1}{2n}$. Also, $p_1'$ is always positive. So we are done with computing $\mathcal{D}_1'$. We can compute $\mathcal{D}_2'$ in the same way. $\qquad\square$

*Proof of Theorem 4.* When $\mathbf{f}$ is strategyproof, it must hold that $f_i(\mathbf{m}) - c_i \cdot m_i = f_i(\mathbf{m}') - c_i \cdot m_i'$ for any two neighboring $\mathbf{m}, \mathbf{m}' \in M$. If $f_i(\mathbf{m}) - c_i \cdot m_i > f_i(\mathbf{m}') - c_i \cdot m_i'$, agent $i$ will misreport their distribution when the ground truth is $(\mathcal{D}_i', \mathcal{D}_{-i})$; else if $f_i(\mathbf{m}) - c_i \cdot m_i < f_i(\mathbf{m}') - c_i \cdot m_i'$, agent $i$ will misreport when the ground truth is $\mathcal{D}$, which conflicts with truthfulness. Since $M$ is connected, we have $f_i(\mathbf{m}) - c_i \cdot m_i = C_i$ for all $\mathbf{m} \in M$. $\qquad\square$

# F  RESULTS FOR EXPECTED ACCURACY OBJECTIVE

## F.1  APPROXIMATION ALGORITHM

Recall that the optimization problem with the expected accuracy objective given an error parameter $\varepsilon$ is

$$\min_{\mathbf{m} \in \mathbb{N}^k} \quad \mathbf{c}^\top \mathbf{m}$$

subject to

$$\max_{h^\star \in \mathcal{H}} \mathbb{E}_{S \sim \mathcal{P}(\mathcal{D}, \mathbf{m}, h^\star)} \left[ \mathrm{err}_{\mathcal{D}_i, h^\star}^{\mathrm{ERM}}(S) \right] \leq \varepsilon, \forall i \in \mathcal{A}\,. \tag{10}$$

Let's denote the optimal solution to the above problem as $\mathbf{m}^{\star, \exp}(\varepsilon)$.

For any pair $(h_j, h_t)$ write

$$E_{j,t} := \Big\{ \text{no sample lies in DIS}(h_j, h_t) \Big\}, \qquad a_{j,t}^i := \mathcal{D}_i\big(\text{DIS}(h_j, h_t)\big),$$

the "no-sample" event and its probability mass under agent $i$'s distribution $\mathcal{D}_i$.

Without loss of generality, relabel the hypotheses so that the disagreement masses are non-increasing:

$$a_1^i \geq a_2^i \geq \ldots \geq a_K^i.$$

Define the first "small" index by

$$n := \min\{ j \mid a_j^i \leq \tfrac{\varepsilon}{2} \} \quad (\text{set } n = K \text{ if no such } j \text{ exists}).$$

$$\Lambda_{<j} := \bigwedge_{k<j} \neg E_k, \qquad \Delta_{\leq j} := \bigvee_{k \leq j} E_k, \qquad a_{K+1}^i := 0.$$

These abbreviations mean, respectively, "no disagreement observed yet" and "some disagreement observed by step $j$."

$$\mathbb{E}_{S \sim \mathcal{P}(\mathcal{D}, m, h^\star)}\Big[ \text{err}_{\mathcal{D}_i, h^\star}^{\text{ERM}}(S) \Big] = \sum_{j=1}^{H-1} \Pr\big(E_j \wedge \Lambda_{<j}\big)\, a_j^i$$

$$\leq \sum_{j=1}^{n} \Pr\big(E_j \wedge \Lambda_{<j}\big)\, a_j^i \ + \ \frac{\varepsilon}{2}$$

$$= \sum_{j=1}^{n} \Pr\big(\Delta_{\leq j}\big)\, \big(a_j^i - a_{j+1}^i\big) \ + \ \frac{\varepsilon}{2}.$$

For any index set $[j] = \{1, \ldots, j\}$ let

$$\Delta_{\leq j} := \bigvee_{t \leq j} E_t.$$

Then

$$\sup_{t \leq j} \Pr(E_t) \ \leq \ \Pr(\Delta_{\leq j}) \ \leq \ \sum_{t=1}^{j} \Pr(E_t) \ \leq \ j \sup_{t \leq j} \Pr(E_t). \tag{UB}$$

Replacing $\Pr(\Delta_{\leq j})$ by its upper bound $\sum_{t=1}^{j} \Pr(E_t)$ in the telescoping sum from the previous step yields

$$\sum_{j=1}^{n} \Big(\sum_{t=1}^{j} \Pr(E_t)\Big)\big(a_j^i - a_{j+1}^i\big) + \frac{\varepsilon}{2}$$

$$= \ \sum_{t=1}^{n} \Pr(E_t)\, a_t^i \ + \ \frac{\varepsilon}{2} \ \leq \ \varepsilon,$$

where we set $a_{n+1}^i = 0$ for the telescoping identity.

Each probability

$$\Pr(E_t) = \prod_{k=1}^{K} \big(1 - \mathcal{D}_k(\text{DIS}(h_0, h_t))\big)^{m_k}$$

is a product of log-affine functions of $\mathbf{m} = (m_1, \ldots, m_K)$ and is therefore convex; the entire left-hand side above is a non-negative weighted sum of convex functions, hence convex in $\mathbf{m}$.

Impose the term-wise bound $\Pr(E_t)\, a_t^i \ \leq \ \frac{\varepsilon}{2H}$ for every disagreement index $t$. Define the set of "large-mass" pairs

$$\mathcal{P}_{\varepsilon/2} := \Big\{ (h_1, h_2) \in \mathcal{H}^2 : \ \max_{i \leq k} \mathcal{D}_i\big(\text{DIS}(h_1, h_2)\big) > \tfrac{\varepsilon}{2} \Big\}.$$

For each such pair write

$$p_i(h_1, h_2) := \mathcal{D}_i(\text{DIS}(h_1, h_2))$$

and

$$a(h_1, h_2) := \min_{i \,:\, p_i(h_1, h_2) > \varepsilon/2} p_i(h_1, h_2).$$

The resulting LP is

$$\min_{\mathbf{m} \in \mathbb{N}^k} \ \mathbf{c}^\top \mathbf{m}$$
$$\text{s.t.} \quad \sum_{i=1}^{k} m_i \, \log\!\big(1 - p_i(h_1, h_2)\big) \ \leq \ \log\!\Big(\tfrac{\varepsilon}{2Ha(h_1, h_2)}\Big), \quad \forall (h_1, h_2) \in \mathcal{P}_{\varepsilon/2}. \tag{11}$$

**Theorem 7.** *The solution to Equation 11 is a feasible solution to Equation 10. Given the optimal solution $\mathbf{m}^{\star, exp}(\tfrac{\varepsilon}{4})$ to Equation 10 with error parameter $\tfrac{\varepsilon}{4}$, then $\log(2H)\mathbf{m}^{\star, exp}(\tfrac{\varepsilon}{4})$ is a feasible solution to Equation 11.*

*Proof of Theorem 7.* It is direct to see the solution to Equation 11 is a feasible solution to Equation 10. Given contribution $\mathbf{m} = \mathbf{m}^{\star,\exp}(\tfrac{\varepsilon}{4})$, we have $\Pr(E_{j,t})a^i_{j,t} \leq \tfrac{\varepsilon}{4}$ for all $h_j, h_t \in \mathcal{H}$. That is to say,

$$\prod_{i'=1}^{k} \left(1 - a^{i'}_{j,t}\right)^{m_{i'}} \leq \frac{\varepsilon}{4a^i_{j,t}}.$$

For $a^i_{j,t} > \tfrac{\varepsilon}{2}$, since $\log(2H) = \log H + 1 \geq \frac{\log(a^i_{j,t} H/\varepsilon)}{\log(4a^i_{j,t}/\varepsilon)} = \frac{\log H + \log(a^i_{j,t}/\varepsilon)}{\log(4a^i_{j,t}/\varepsilon)}$, we have

$$\prod_{i'=1}^{k} \left(1 - a^{i'}_{j,t}\right)^{m_{i'} \cdot \log(2H)} \leq \prod_{i'=1}^{k} \left(1 - a^{i'}_{j,t}\right)^{m_{i'} \cdot \frac{\log(a^i_{j,t} H/\varepsilon)}{\log(4a^i_{j,t}/\varepsilon)}} \leq \frac{\varepsilon}{a^i_{j,t} H}.$$

$\square$

**The performance of running ERM over the cover $\overline{\mathcal{H}}$ for the expected accuracy objective.** For any $\mathcal{H}$, let $\overline{\mathcal{H}}$ be a $\gamma$-cover of $\mathcal{H}$. Then for any contribution $\mathbf{m}$ satisfying this constraint, any $h^\star \in \mathcal{H}$ and any agent $i$, given the labeled data $S^{h^\star} = \cup_{j\in[k]}\{(x, h^\star(x))|x \in U_j\}$, let $\overline{\text{err}} = \max\{\text{err}_{\mathcal{D}_i}(\overline{h})|\overline{h} \in \overline{\mathcal{H}} : \text{err}_{S^{h^\star}}(\overline{h}) = \min_{h' \in \overline{\mathcal{H}}} \text{err}_{S^{h^\star}}(h')\}$ denote the worst case error when running ERM over $\overline{\mathcal{H}}$. We can show that $\overline{\text{err}}$ is small in expectation. More specifically, when we can achieve $\varepsilon/2$-expected accuracy guarantee for $\overline{\mathcal{H}}$, we have
When we can achieve $\varepsilon$ expected accuracy guarantee for the cover $\overline{\mathcal{H}}$, we have

$$\mathbb{E}\left[\overline{\text{err}}\right] \leq \mathbb{E}\left[\overline{\text{err}} \,\Big|\, \text{err}_{S^{h^\star}}(\overline{h}^\star) = 0\right] \mathbb{P}\left(\text{err}_{S^{h^\star}}(\overline{h}^\star) = 0\right) + \mathbb{P}\left(\text{err}_{S^{h^\star}}(\overline{h}^\star) \neq 0\right)$$
$$\leq \mathbb{E}\left[\overline{\text{err}} \,\Big|\, \text{err}_{S^{h^\star}}(\overline{h}^\star) = 0\right] + \mathbb{P}\left(\text{err}_{S^{h^\star}}(\overline{h}^\star) \neq 0\right)$$
$$\leq \varepsilon/2 + \left(1 - (1-\gamma)^{\frac{k\text{VCdim}(\mathcal{H})}{\varepsilon}}\right).$$

By setting $\gamma = O(\frac{\varepsilon^2}{k \cdot \text{VCdim}(\mathcal{H})})$, we have $\mathbb{E}\left[\overline{\text{err}}\right] \leq \varepsilon$.

### F.2 LOCAL OBLIVIOUSNESS OF THE APPROXIMATION ALGORITHM

Note that the image space of this approximation algorithm is $[0, \frac{2\log(H)}{\varepsilon}]^k$ according to Equation 11. Then we show that most area of $[0, \frac{2\log(H)}{\varepsilon}]^k$ is oblivious.

**Lemma 5.** *For any $k < H \in \mathbb{N}$ and $\varepsilon < 1 - \frac{1}{2H}$, there exists an instance of expected accuracy learning instance of $(\mathcal{H}, \varepsilon)$ with $|\mathcal{H}| = H$ in the $k$-agent setting such that the approximation algorithm introduced in Theorem 7 is oblivious at all $\mathbf{m} \in [2 + \frac{\log(1/\varepsilon)}{\log(2H)}, \frac{2\log(H)}{\varepsilon}]^k$.*

*Proof.* Let the input space be the $H$ points $\mathcal{X} = \{x_1, \ldots, x_H\}$. The hypothesis class $\mathcal{H}$ consists of all singletons over $\{x_1, \ldots, x_{H-1}\}$—denote them $h_1, \ldots, h_{H-1}$—together with the all-negative

hypothesis $h_0$. Fix a contribution vector $\mathbf{m} = (m_1, \ldots, m_k)$. For each agent $i \in [k]$ choose $c_i \in [\varepsilon/2, \ 1 - \frac{1}{2H}]$ that solves

$$(1 - c_i)^{m_i} c_i \ = \ \frac{\varepsilon}{2H}. \tag{1}$$

Such a solution exists whenever

$$m_i \ \in \ \big[\, 1 + \tfrac{\log(1/\varepsilon)}{\log(2H)}, \ \tfrac{2 \log H}{\varepsilon} \,\big].$$

Define $\mathcal{D}_i$ by setting $\mathcal{D}_i(x_i) = c_i$ and $\mathcal{D}_i(x_H) = 1 - c_i$.

Fix an agent $i$ and a hypothesis $h_j$ with $j \in [H - 1]$. Under $h_j$ the expected ERM error of agent $i$ equals

$$\mathbb{E}_{S \sim \mathcal{P}(\mathbf{\mathcal{D}}, \mathbf{m}, h_j)}\big[\mathrm{err}^{\mathrm{ERM}}_{\mathcal{D}_i, h_j}(S)\big] = \Pr[x_i \text{ and } x_j \text{ both unseen in } S]\, c_i \ \leq \ \Pr[x_i \text{ unseen in } S]\, c_i.$$

For the all-negative hypothesis $h_0$ the same bound holds:

$$\mathbb{E}_{S \sim \mathcal{P}(\mathbf{\mathcal{D}}, \mathbf{m}, h_0)}\big[\mathrm{err}^{\mathrm{ERM}}_{\mathcal{D}_i, h_0}(S)\big] = \Pr[x_i \text{ unseen in } S]\, c_i.$$

Because $c_i \leq 1 - \frac{1}{2H}$, if agent $i$ contributes only $m_i - 1$ samples then

$$\Pr[x_i \text{ unseen in } S]\, c_i = (1 - c_i)^{m_i - 1} c_i \ \leq \ \frac{\varepsilon}{2H(1 - c_i)} \ \leq \ \varepsilon.$$

Hence $\mathbf{m} - \mathbf{1}$ is also feasible for the profile $\mathbf{\mathcal{D}}$. More generally, any Hamming neighbour $\mathbf{m}'$ of $\mathbf{m}$ is feasible for some modified profile $(\mathcal{D}'_i, \mathbf{\mathcal{D}}_{-i})$, so the approximation algorithm remains oblivious throughout the specified hyper-rectangle.

$\square$

