# OpenReview forum: "Incentives in Federated Learning with Heterogeneous Agents"
_ICLR.cc/2026/Conference — ICLR 2026 Poster_

### Official Review · Reviewer_4jxP · 2025-10-24

**Soundness:** 3
**Presentation:** 3
**Contribution:** 3
**Rating:** 6
**Confidence:** 4

**Summary:**

This paper studies incentive alignment in federated learning (FL) with heterogeneous data distributions. Unlike prior work assuming data exchangeability, agents' utilities depend on *who* contributes samples. The authors show that uncoordinated play yields unbounded Price of Stability, prove that computing optimal allocations is NP-hard, design a polynomial-time LP approximation with logarithmic guarantees, and propose a strategyproof "pay-what-you-contribute" mechanism that is essentially unique within contribution-based transfers.

**Strengths:**

1. **Novel heterogeneity model:** Properly captures that agents benefit more from their own data than others' (validated empirically in Figure 1), addressing a key gap in FL incentive literature that assumed exchangeable data.
2. **Comprehensive theoretical analysis:** Rigorous results spanning game theory (unbounded PoS), computational complexity (NP-hardness), approximation algorithms (log-factor LP), and mechanism design (uniqueness of PWYC). The progression from negative to positive results is well-structured.
3. **Clean technical contributions:** The Set Cover reduction (Theorem 2) is elegant, the LP analysis properly handles infinite hypothesis classes via VC theory, and the mechanism uniqueness result (Theorem 4) provides interesting theoretical justification for simple mechanisms.
4. **Clear presentation:** Well-motivated with good intuitive examples (warm-up in Section 2.2, two-player illustration in Section 3) before formal results.

**Weaknesses:**

1. **Restrictive modeling assumptions:** Binary utilities (agents only care about meeting thresholds, not exceeding them) and the need for external subsidies in PWYC limit practical applicability. These limitations deserve more prominent discussion in the main text, not just Appendix B.
2. **Weak empirical validation:** Experiments relegated to appendix with small hypothesis classes (|H| ≤ 50) and limited datasets. For ICLR, stronger validation on realistic FL scenarios is expected. Scalability to neural networks unclear.
3. **Loose approximation bounds:** The O(d²(log(...))²/log(1/δ)) guarantee for infinite H may be impractical. No discussion of when this is acceptable or tighter analysis for special cases. No lower bounds showing necessity of logarithmic factors.
4. **Limited practical guidance:** Unclear how to implement the LP allocation for neural network hypothesis classes. The transition from known distributions (Section 4) to unknown (Section 5) could be smoother.

**Questions:**

1. Can you provide lower bounds showing the logarithmic approximation is necessary, or identify problem classes admitting better approximations?
2. Can you formally characterize when budget-balanced mechanisms exist in your setting, or prove impossibility?
3. How would the LP approach work for neural network hypothesis classes in practice?
4. How do results extend to heterogeneous accuracy requirements (different ε_i) or sequential/multi-round FL?
5. Can you provide concrete payment comparisons between PWYC and approximate VCG mechanisms?

---

> ### Author Response · Authors · 2025-11-19
>
> > Restrictive modeling assumptions: Binary utilities (agents only care about meeting thresholds, not exceeding them) and the need for external subsidies in PWYC limit practical applicability.
>
>
> This is a fair point. In the revision, we now highlight this argument more prominently in the main text (lines 223–238) and in the updated Appendix B.
>
> **On the utility side**, we emphasize that the binary threshold model is a modeling choice, not a literal claim that agents have zero utility below the target. This payoff can represent a broad class of quasi-linear utilities and captures richer behavior than its simple indicator form suggests at first glance.
>
>   Our agent-level problem is “minimize own cost subject to achieving a required accuracy.” Appendix B shows, via a Lagrange-multiplier argument, that any optimal solution to this constrained problem can also be seen as optimal for an unconstrained objective that trades off accuracy against cost: there exists a multiplier $\lambda_i$ such that any optimum under the accuracy constraint is also optimal for a Lagrangian objective that values accuracy and penalizes cost at rate $\lambda_i$. Thus the stopping point for a satisficing agent in our threshold model coincides with the optimal trade-off point for a richer quasi-linear utility. Therefore, ex post, using a threshold utility does not exclude the optimal outcome that a more complex increasing utility would select.
>
> **On the PWYC mechanism side**, we note (lines 439–449 in the revised version) that while it is not budget-balanced, this limitation is common in mechanism design (e.g., the celebrated VCG mechanism also requires external payments). That said, PWYC does not necessarily require the planner to bear the full cost (see our answer below).
>
> ### Questions:
> > 1. Can you provide lower bounds showing the logarithmic approximation is necessary, or identify problem classes admitting better approximations?
>
> Our mechanism-design results in Section 5 apply to any approximation algorithm satisfying the conditions in Definition 1. For a given hypothesis class $\mathcal H$, any approximation algorithm that better exploits structure or uses data-dependent estimates can be plugged into our framework.
>
> For the specific approximation algorithm in Theorem 3, the proof (Appendix D.2) suggests a logarithmic overhead is necessary: we enforce all agents’ accuracy constraints via a union bound over all $|\mathcal H|$ “bad” hypotheses, which introduces a $\log(1/\delta)$ factor that is tight when these bad events are essentially independent. Empirically, our FEMNIST validation (Appendix A.2) shows that the LP allocation incurs only a modest overhead.
>
>
>
> > 2. Can you formally characterize when budget-balanced mechanisms exist in your setting, or prove impossibility?
>
> We agree that understanding the (im)possibility of budget-balanced mechanisms in our setting is important. Our results indicate that exact budget-balance and strategyproofness cannot be achieved simultaneously. We show that within the class of contribution-based, locally-oblivious transfers, PWYC is the unique strategyproof mechanism, which is not budget-balanced.
>
> This parallels classic public-goods settings (e.g., VCG), where dominant-strategy truthfulness and exact budget-balance are incompatible and subsidies are unavoidable. In our case, PWYC runs a deficit, but the constants $C_i$ can be tuned to recover part of the cost without breaking strategyproofness, so the planner need not pay the entire cost.
>
>
> > 3. How would the LP approach work for neural network hypothesis classes in practice?
>
> Our LP analysis in Section 4 applies in principle to rich hypothesis classes such as neural networks, by working with VC theory rather than enumerating hypotheses. Importantly, though, the PWYC mechanism applies to any approximation algorithm satisfying Definition 1, so in practice one can substitute empirical or data-dependent allocation procedures tailored to the model class at hand, without changing our qualitative incentive conclusions.
>
>
>
> > 4. How do results extend to heterogeneous accuracy requirements (different ε_i) or sequential/multi-round FL?
>
> Our entire framework – including each agent’s utility function, the equilibrium analysis, the LP-based optimal allocation, and the mechanism design – generalizes directly to heterogeneous accuracy thresholds (i.e. different $\varepsilon_i$), as mentioned in footnote 2 and Section 6.
>
> Regarding sequential or multi-round FL, the key point is that these protocols specify how the total sample counts are realized over time, while our objective only cares about the resulting _distributions_ rather than round-by-round trajectories.
>
>
> > 5. Can you provide concrete payment comparisons between PWYC and approximate VCG mechanisms?
>
>
> Could you clarify what kind of “concrete payment comparisons” you have in mind between PWYC and approximate VCG?

---

### Official Review · Reviewer_hAX9 · 2025-11-02

**Soundness:** 3
**Presentation:** 3
**Contribution:** 2
**Rating:** 8
**Confidence:** 3

**Summary:**

This paper studied a game-theoretic framework in federated learning with heterogeneous data, where an agent’s utility depends on the specific samples that each agent supplies. The agents aim to meet a PAC-style accuracy threshold while minimizing cost. Theoretically, first, the paper showed that without coordination the game can fail badly and pure equilibria is not guaranteed to exist. Second, with coorporation, it is NP-hard to obtain the cost-minimizing contribution vector. The paper then developed a polynomial-time linear program that achieves a logarithmic approximation. Lastly, the paper proposed a mechanism with payment rule that is strategy-proof and unique within the class of contribution-based transfers.

**Strengths:**

Overall I found this work to be interesting and well-presented, and it provided a useful set of threotical analysis and results which extended upon the current literature on incentives in FL with heterogeneous data.
- Theoretically, this paper introduced a concrete utility model based on PAC-style threshold guarantees. And such utility model allowed further analysis on the necessity of coordination and the payment-based mechanism. While most of the existing work did not use specific utility models, this framework could be a good extension on those.
- the strategyproof mechanism used a simple pay-what-you-contribute rule that is natural and intuitive
- while the analysis assumed linear cost functions for agents, the results are flexible to hold with more general convext cost functions

**Weaknesses:**

- I found the utility function assumed for the agents to be strict, in particular the payoff is an indicator function that's either 1 or 0. In reality the payoff to agent may be a more general increasing function of the number of samples, such as (1-loss) illustrated in Figure 1. Can the results be extended to more general utility funcitons?

**Questions:**

Results in section 5 assumed that the planner knows c_i. In practice it's more likely that the agents also report their cost function c_i. Is it possible to have a strategyproof mechanism where the agents report both distributions and their costs?

---

> ### Author Response · Authors · 2025-11-19
>
> > I found the utility function assumed for the agents to be strict, in particular the payoff is an indicator function that's either 1 or 0. In reality the payoff to agent may be a more general increasing function of the number of samples, such as (1-loss) illustrated in Figure 1. Can the results be extended to more general utility funcitons?
>
>
> This is a fair concern and closely related to Reviewer qTMm’s point. As noted in the Discussion, we use the binary “meets target or not” payoff as a modeling choice, not as a claim that agents derive literally zero benefit until the target is met.
>
> Importantly, though, we argue that the binary threshold payoff captures a richer class of utility functions than its simple indicator form suggests at first glance. Our agent-level decision problem is “minimize own cost subject to achieving a required accuracy.” Appendix B shows that any optimal solution to this constrained problem can also be obtained as an optimal solution of an unconstrained objective that trades off accuracy against cost. In particular, there exists a multiplier $\lambda_i$ such that any optimum under the accuracy constraint is also optimal for a Lagrangian objective that values accuracy and penalizes cost at rate $\lambda_i$. This gives a precise equivalence: the stopping point for a satisficing agent in our threshold model coincides with the optimal trade-off point for a quasi-linear utility that values accuracy and penalizes cost at rate $\lambda_i$. Therefore, ex post, using a threshold utility does not exclude the optimal outcome that a more complex utility would produce.
>
> Put differently, this means we could assign a value $\lambda_i$ per unit of accuracy so that an agent with a continuous utility (for example, a concave accuracy–benefit curve) would stop contributing at the same point where it would just meet an accuracy threshold under the binary model. In that sense, any agent with a smooth, monotone benefit of accuracy and convex cost behaves as if it had a personal threshold (if the utility was not smooth or had multiple local optima, the argument would be more delicate.)
>
>
> > Results in section 5 assumed that the planner knows c_i. In practice it's more likely that the agents also report their cost function c_i. Is it possible to have a strategyproof mechanism where the agents report both distributions and their costs?
>
>
> This is a natural concern, and we agree it is worth making the modeling choice explicit (lines 383-384 in the new revision and footnote 6). For many applications, it is reasonable to treat per-sample costs as known or contractually specified: a platform may post a fixed price per labeled example, meter and bill bandwidth or compute usage, or sign annotation contracts that set a per-label rate up front, so that $c_i$ is a parameter of the environment rather than private strategic information.
>
> If agents were also allowed to report $c_i$ and we continued to reimburse reported cost as in PWYC, they would indeed have an incentive to inflate $c_i$. More broadly, in our public-good setting it is  difficult to design a mechanism that is simultaneously strategyproof in both distributions and costs, individually rational, and not subject to unbounded deficit. Designing a mechanism that truthfully elicits both distributions and costs is therefore a challenging separate problem that we do not attempt to solve here.

---

> > ### Comment · Reviewer_hAX9 · 2025-11-27
> > **acknowledgement of rebuttal**
> >
> > Thanks for the further clarifications on my concerns. I acknowledge that I have read through the author response and taken that into considerations for my evaluations.

---

### Official Review · Reviewer_gnyW · 2025-11-02

**Soundness:** 2
**Presentation:** 3
**Contribution:** 2
**Rating:** 4
**Confidence:** 3

**Summary:**

The paper models federated learning with heterogeneous agents as a contribution game in which each agent wants a PAC-style guarantee on its own distribution Utilities depend on who supplies each sample, not just the total mass. The authors show that uncoordinated play can be pathological: pure NE may not exist and even the best equilibrium can be arbitrarily worse than cooperation. A central planner seeking the cost-minimizing contribution vector faces an NP-hard decision problem, but a linear program delivers logarithmic-type approximation. Coupling this LP with a "pay-what-you-contribute" transfer is strategy-proof and, within contribution-based mechanisms, essentially unique.

**Strengths:**

- Clear formalization of heterogenous valuations in FL where benefits depend on contributor identity; the PoS lower bound and the local-obliviousness–based uniqueness of PWYC are novel angles versus exchangeable-data models.
- Connects learning theory, game theory, and mechanism design for realistic FL, offering a tractable planner and a strategy-proof payment rule that could inform practice.
- Experients small but targeted, including FEMNIST/Shakespeare experiments that validate the heterogeneity premise.

**Weaknesses:**

-  The analysis assumes realizable PAC learning and ERM and requires pairwise disagreement masses for (potentially many) hypothesis pairs. For infinite $\mathcal{H}$ LP, a $\lambda$-net is constructed using the VC dimension. Estimating these quantities reliably for modern models is nontrivial and makes the linear programming approximation intractable.
- PWYC is strategy-proof but typically not budget-balanced. There is no bound on the planner’s deficit or a characterization of near–budget balance vs. individual rationality. Adding such bounds would strengthen the practical message.
- The uniqueness of contribution-based payments is proved for two agents under local obliviousness; it is unclear how far this extends to k>2 or to other approximation algorithms.

**Questions:**

- In theorem 3, why is the approximation error independent on $\epsilon$ for finite $\mathcal{H}$?
- If an agent can split into multiple identities or collude, does PWYC remain strategy-proof and does the LP allocation change in harmful ways?
- What are examples of practical FL protocols that satisfies the information condition in section 5?

---

> ### Author Response · Authors · 2025-11-19
>
> > The analysis assumes realizable PAC learning and ERM and requires pairwise disagreement masses for (potentially many) hypothesis pairs.
>
> We agree that naively requiring pairwise disagreement masses for many hypothesis pairs would be nontrivial at scale. In our analysis (Section 4.2 and Appendix D.2), we approximate an infinite class by a finite cover and solve the allocation on that surrogate, so the guarantees require only a valid cover, not the exact VC dimension. In practice one can use a conservative upper bound or work with a finite surrogate, as we do in our experiment (Appendix A.2). This may over-provision samples, but does not change the qualitative conclusions.
>
> We acknowledge that pure realizability is restrictive as modern models often operate in agnostic settings with label noise or slight model misspecification. However, our results do not hinge on realizability in an overly fragile way. The game-theoretic results in Section 3 and the NP-hardness in Section 4 only require that each agent has a well-defined accuracy constraint. On the mechanism-design side, Section 5 is stated for the broader class of approximation algorithms in Definition 1, not just our specific LP. For a concrete hypothesis class $\mathcal H$, including modern neural models, one can plug in an ad-hoc allocation or approximation scheme that better exploits architectural or domain structure, and our incentive and uniqueness guarantees continue to apply as long as that scheme satisfies the conditions in Definition 1.
>
> > PWYC is strategy-proof but typically not budget-balanced. There is no bound on the planner’s deficit or a characterization of near–budget balance vs. individual rationality.
>
> While we did not frame it explicitly as a “deficit bound,” our analysis already implies a simple one. Under PWYC, the total payment made by the planner is exactly $\sum_i (c_i m_i + C_i)$. Generally, in our model the planner already chooses contributions to (approximately) minimize this cost, so the deficit is tied directly to the cooperative cost required to meet all agents’ accuracy targets, rather than to some uncontrolled quantity.
>
> The constant terms in PWYC give a simple knob to trade off deficit against agent surplus. Individual rationality only requires $u_i \ge 0$, i.e. $C_i \ge -1$, so the planner can uniformly lower these constants (down to the point where agents are just indifferent) and thereby reduce total payments by up to one unit per agent while still keeping everyone willing to participate.
>
> > The uniqueness of contribution-based payments is proved for two agents under local obliviousness; it is unclear how far this extends to k>2 or to other approximation algorithms.
>
> Our uniqueness result has a general and a concrete part. The characterization in Theorem 4 applies to any number of agents and any contribution-based payment rule whose approximation algorithm satisfies Definition 1. The proof of Theorem 4 does not rely on having exactly two agents or on any special property of our LP. The only place we restrict to two agents is Lemma 1, where we verify that our PAC-based LP satisfies Definition 1. This lemma simply shows the assumption is non-vacuous. For other approximation algorithms or for $k>2$, the same uniqueness conclusion holds whenever one can verify Definition 1.
>
> ### Questions
> > In theorem 3, why is the approximation error independent on $\varepsilon$ for finite $\mathcal{H}$?
>
> In Theorem 3, $\varepsilon$ affects the absolute sample size but not the relative approximation error. As $\varepsilon$ decreases, the PAC constraints require that every hypothesis with error $>\varepsilon$ be eliminated with high probability, so any feasible contribution vector (including the optimum $m$ and the LP solution $m^\ast$) must scale its total sample size. The gap comes only from the union bound, which contributes a $\log|\mathcal{H}|$ factor that depends on $|\mathcal{H}|$ and $\delta$ but not on $\varepsilon$, so $|m|$ and $|m^\ast|$ both blow up as $\varepsilon \to 0$ but their ratio does not.
>
> > If an agent can split into multiple identities or collude, does PWYC remain strategy-proof and does the LP allocation change in harmful ways?
>
> PWYC is strategyproof under the standard one-identity-per-agent assumption. If agents can split or collude, this proof no longer directly applies. That said, if after the split the clones truthfully report their shared distribution, the optimal contribution vector and LP solution are unchanged. No special harm comes to allocation or fairness, beyond possibly exploiting fixed per-agent payments (the constants $C_i$), which can be neutralized.
>
> > What are examples of practical FL protocols that satisfies the information condition in section 5?
>
> Could you clarify what you mean by the information condition in Section 5? Do you have in mind protocols where each agent shares summary statistics instead of raw data, or approximation rules that satisfy local obliviousness?

---

### Official Review · Reviewer_qTMm · 2025-11-03

**Soundness:** 3
**Presentation:** 2
**Contribution:** 2
**Rating:** 4
**Confidence:** 3

**Summary:**

This paper uses the framework of game theory to study the incentives of agents in federated learning to contribute local data in a heterogeneous learning setup.
It models the utility functions of agents by assuming that each agent has a PAC accuracy objective—achieving at most $\varepsilon$ empirical risk with probability at least $1-\delta$—which yields a payoff of 1 if and only if the goal is achieved, while collecting and contributing data incurs a linear cost.
Based on this formulation, the paper shows that a non-coordinated equilibrium may not exist or can be arbitrarily inefficient compared to the coordinated social optimum.
It proves that computing the social optimum is NP-hard but provides a scheme to approximate it via linear programming (LP).
Finally, it proposes the Pay-What-You-Contribute (PWYC) scheme, where each agent is paid according to its data-collection cost, which is strategyproof in the sense that agents have no incentive to misreport their local data.

**Strengths:**

The study of incentivizing agents to participate in collaborative federated learning and preventing free-riding is of clear interest to the community. The paper is generally well-written, and the conceptual takeaways of each section are fairly clear and interesting. It provides a good theoretical prototype for studying agents’ social behavior in fully heterogeneous federated learning settings.

**Weaknesses:**

My key concern is that although the high-level ideas are compelling, the analysis relies on multiple layers of simplifications and idealized assumptions, which makes the overall significance of this work questionable.

**1. Assumption on sample collection.**
Section 2.1 explicitly requires that the local datasets should be pooled into a collective dataset $S$, and that the global model is directly trained via ERM on $S$. To my understanding, this does not conform to the philosophy of federated learning, which aims to keep local data private. In practice, algorithms such as FedAvg are used to indirectly train a central model (hypothesis) over $S$, which introduces a gap between what the federated learning mechanism can achieve and the true minimizer of ERM. The paper seems to ignore this gap and does not explain or justify this simplification.

**2. Binary payoff structure.**
It is assumed that the payoff from model training is given by the binary variable $a_i^{\epsilon,\delta}$, indicating whether the PAC learning objective is satisfied or not. This is an unrealistic assumption. One might argue that such a simplification is necessary for analytical tractability, but it is not merely a matter of convenience. For instance, I do not see a straightforward way to generalize notions such as Assumption 1 (self-sufficiency) or the socially optimal data-contribution profile to the case of non-binary payoffs.

**3. Requirement to fix $m_i$ in advance.**
This is another stringent and unrealistic assumption, as in practice, local model optimization can be performed independently, and the utilization of local data should be dynamically adjustable. It is also unclear how the central sample-allocation scheme would operate when the hypothesis space’s covering number or VC dimension, as well as the agents’ explicit data-collection costs, are unknown. The framework quickly becomes impractical due to $m_i$ being pre-fixed, especially when it becomes difficult to approximate these quantities accurately.

**4. Is PWYC making federated learning meaningless?** I am uncertain about the implications of the results in Section 5. The PWYC mechanism simply reimburses each agent for its data-collection cost, meaning that the central server essentially subsidizes all data acquisition. However, if the server pays for everything, this seems to undermine the motivation for federated learning in the first place, as it assumes the existence of an entity that can simply “buy all data” and conduct centralized training. The result in Section 5.2 is a bit disappointing in this regard, as its message appears to be that this is essentially the only case where agents can be sufficiently incentivized to participate in federated learning without misreporting data.

**Questions:**

Please see the Weaknesses section.

---

> ### Author Response · Authors · 2025-11-19
> **Part 1/2**
>
> We thank the reviewer for the detailed comments. Due to the 5,000-character limit, this reply is split into two parts.
>
>
>
> > 1. Assumption on sample collection. Section 2.1 explicitly requires that the local datasets should be pooled into a collective dataset $S$, and that the global model is directly trained via ERM on $S$. To my understanding, this does not conform to the philosophy of federated learning, which aims to keep local data private.
>
>
> This is a good point. Pooling all local datasets and training a centralized ERM would indeed undermine a key benefit of FL, namely that raw data remains on users’ devices and is never aggregated on the server. In practice, algorithms such as FedAvg can approximate the pooled-ERM solution through local updates and model aggregation, without ever constructing a pooled dataset.
>
> However, **ERM on a pooled dataset $S$ is a theoretical reference point, not an implementation assumption:** it defines the PAC constraints and certifies that, if one *were* to train on the pooled draw of size $m$, the resulting model would satisfy each agent’s $(\varepsilon_i,\delta)$ requirement. This lets us abstract away optimization dynamics and focus on incentives and sample complexity, while any FL protocol that gets sufficiently close to this benchmark inherits the same qualitative guarantees. That is to say, ERM on $S$ can be implemented by FedAvg without pooling local datasets and doesn't conflict with the philosophy of keeping local data private.  This use of centralized ERM as an upper-bound reference has also been adopted in prior work [1–3].
>
>
> We have revised the manuscript to incorporate this in the Discussion section (starting at line 513), in Section 2 (Footnote 1), and in Section 4 (starting at line 343).
>
> [1] Li et al. Analyzes the FedAvg algorithm by comparing it to the global ERM solution, explicitly using the pooled-data optimum as a reference for convergence. 2020.
>
> [2] Mohri et al. Agnostic Federated Learning. ICML 2019.
>
> [3] Avrim Blum et al. One for one, or all for all: Equilibria and optimality of collaboration in federated learning. ICML 2021.
>
>
> > 2. Binary payoff structure. It is assumed that the payoff from model training is given by the binary variable , indicating whether the PAC learning objective is satisfied or not. This is an unrealistic assumption. One might argue that such a simplification is necessary for analytical tractability, but it is not merely a matter of convenience.
>
>
>
> This is a fair concern. We use the binary “meets target or not” payoff as a modeling choice, not as a claim that agents derive literally zero benefit until the target is met.  We argue that **the binary threshold payoff captures a richer class of utility functions than its simple indicator form suggests** at first glance.
>
> Our agent-level decision problem is “minimize own cost subject to achieving a required accuracy.” The newly revised Appendix B shows that any optimal solution to this constrained problem can also be obtained as an optimal solution of an unconstrained objective that trades off accuracy against cost. In particular, there exists a multiplier $\lambda_i$ such that any optimum under the accuracy constraint is also optimal for a Lagrangian objective that values accuracy and penalizes cost at rate $\lambda_i$.
>
> Put differently, the stopping point for a satisficing agent in our threshold model coincides with the optimal trade-off point for a quasi-linear utility that values accuracy and penalizes cost, so ex post, using a threshold utility does not exclude the optimal outcome that a more complex utility would produce. We have revised the paper to make this argument more pronounced (lines 223–238 and an updated Appendix B).
>
> We also note that threshold-based formulations have been used in prior work as well [3,4].
>
> [3] Avrim Blum et al. One for one, or all for all: Equilibria and optimality of collaboration in federated learning. ICML 2021.
>
> [4] Avrim Blum et al. Collaborative PAC learning. NeurIPS 2017.
>
> >  For instance, I do not see a straightforward way to generalize notions such as Assumption 1 (self-sufficiency) or the socially optimal data-contribution profile to the case of non-binary payoffs.
>
> Assumption 1 is meant to encode that each agent has a meaningful fallback (obtain **positive utility** by training alone). For more general increasing utilities, the analogous condition would simply be that there exists a standalone contribution giving nonnegative utility without collaboration. Likewise, the socially optimal data-contribution profile is defined as the contribution vector that **maximizes total welfare** (in our model this reduces to the LP objective in Equation (4)).

---

> ### Author Response · Authors · 2025-11-19
> **Part 2/2**
>
> > 3. Requirement to fix $m_i$ advance. This is another stringent and unrealistic assumption, as in practice, local model optimization can be performed independently, and the utilization of local data should be dynamically adjustable.
>
> $m$ is computed by the planner or the mechanism and "pre-fixing” means an _ex ante commitment_ to a contribution cap: each agent agrees to use up to $m_i$ local samples in the collaborative training process, rather than adjusting its data usage on the fly (e.g. via online learning, active sampling, or an adaptive stopping rule). In our model, the objective is distributional (PAC‑style or expected‑error) and does not depend on a particular training trajectory. In practice, one can implement our allocation by treating $m_i$ as a budget on how many local examples agent $i$ may use in the protocol (for instance, by fixing a committed subset of size $m_i$ at the outset).
>
>
> > It is also unclear how the central sample-allocation scheme would operate when the hypothesis space’s covering number or VC dimension, as well as the agents’ explicit data-collection costs, are unknown. The framework quickly becomes impractical due to $m_i$ being pre-fixed, especially when it becomes difficult to approximate these quantities accurately.
>
>
> **VC dimension and covers**. Requiring a bound on hypothesis-class complexity is standard in PAC analyses. In our analysis (Section 4.2 and Appendix D.2), we approximate an infinite class by a finite cover and solve the allocation on that surrogate; the guarantees depend only on having a valid cover, not on knowing the exact VC value. In practice, one can use a conservative upper bound on capacity (for example, based on the model architecture) or work with a finite surrogate hypothesis set, as we do in our FEMNIST experiment (Appendix A.2). This may over-provision samples, but does not change the qualitative conclusions. Estimating or upper-bounding such complexity parameters has been studied in the literature [5–6].
>
> **Per‑sample costs**. In our analysis we treat $c_i$ as known or contractible. In many applications, agents do have a clear notion of their per-sample cost. These costs can be estimated or agreed upon in advance (e.g. a platform might set a standard reimbursement rate per data point).
>
>
> [5] Shao et al. Measuring the VC-dimension using optimized experimental design. Neural Computation 2020.
>
> [6] Aslan et al. Calculating the VC-dimension of decision trees. International Symposium on Computer and Information Sciences 2009.
>
> > 4. Is PWYC making federated learning meaningless? I am uncertain about the implications of the results in Section 5. The PWYC mechanism simply reimburses each agent for its data-collection cost, meaning that the central server essentially subsidizes all data acquisition. However, if the server pays for everything, this seems to undermine the motivation for federated learning in the first place, as it assumes the existence of an entity that can simply “buy all data” and conduct centralized training. The result in Section 5.2 is a bit disappointing in this regard, as its message appears to be that this is essentially the only case where agents can be sufficiently incentivized to participate in federated learning without misreporting data.
>
> At first glance, if the central server reimburses all data-collection costs, it can look as if the server is simply “buying all the data,” which understandably raises the concern that this could undercut the original motivation for federated learning.
>
> We view the uniqueness of PWYC as a substantive, if somewhat negative, insight: if one insists on dominant-strategy truthfulness and approximate efficiency in a heterogeneous FL game, then some external budget or cost-sharing is unavoidable—much as in other public-goods settings where reimbursement is often necessary to align incentives. This does not make federated learning meaningless. It clarifies what a realistic FL system with self-interested agents must do to unlock gains without centralizing data. PWYC adds the incentive layer: it fairly compensates contributions so agents participate and report honestly. When a coordinator is willing to cover costs (as is often implicitly the case), one gets the best of both worlds—decentralized training with truthful, sufficient contributions.
>
> As we note (line 447 in the revised version), PWYC does not force the planner to shoulder all costs: payments are $p_i(m)=c_i m_i + C_i$, and the constants $C_i$ can be set to recover part of the spend from participants or downstream users while preserving strategyproofness.

---

### Meta-Review · Area_Chair_cCAy · 2026-01-01

**Summary:**

This paper provides a neat theoretical formulation capturing some of the intricacies of incentives in FL.
 - captures heterogeneous data distributions and its effects
 - demonstrates co-ordination failure arising due to "threshold" utilities
 - provides polynomial-time approximation algorithms with provable guarantees to compute the central optimum
 - analyzes auction mechanisms when local distribution is private.

However there were significant limitations raised. While most reviewers would likely recommend that this work be accepted despite these, reviewer gnyW might have differed. While gnyW's concerns are all valid, they are not unique to this paper and will hopefully be addressed in future work and extensions.

Having said this, i fully concur with all the reviewers that the limitations of the framework and results were not sufficiently discussed. While I recommend acceptance, I expect the authors to **incorporate the discussions on the limitations raised in the reviewer concerns below**.

**Reviewer Concerns:**

- Reviewer qTMm raised questions all dealing with how the theoretical results in this paper would translate into real world FL.
    - need for centralized collection - fully addressed in rebuttal
    - restriction to binary utility - rebuttal argues that binary utilities can be seen as constraints and have a one to one mapping to quasi-linear utilities (via arguments similar to strong duality / lagrange relaxation). This argument only partially addresses the concern - it would have been much better if the authors were more formal with their App B e.g. by having a concrete corollary for the general case.
    - $m_i$ fixed - addressed
    - knowledge of unknown paramters by the central planner - rebuttal provides some workarounds but this is a fundamental limitation of current work
    - practicality of PWYC - addressed

- Reviewer gnyW concerns:
    - realizability assumption - not alleviated but rebuttal argues some of their arguments might generalize beyond this.
    - budget balancedness - not addressed. Reviewer was likely looking for a bound in terms of parameters defined in problem (e.g. $\epsilon, \delta, n, c, \dots$) rather than post-hoc $m_i$ or $C_i$.
    - uniqueness if payment rule - addressed
    - question about theorem result - addressed
    - strategyproofness under splitting - acknowledge weakness
    - practical implementation - not addressed. Reviewer likely meant how can one implement step 1. in the protocol where agents share their distribution. In real settings there is no succint representation of this and is further likely unknown to the agents.


- Reviewer hAX9:
    - restriction to binary utility - partially addresed (see qTMm)
    - unknown $c_i$ - rebuttal provides some workarounds but this remains a fundamental limitation. Double auctions could have been studied for this.

- Reviewer 4jxP overall notes that the limitations need to be conveyed more explicitly
    - restriction to binary utility - partially addresed (see qTMm)
    - budget balancedness - provides partial workarounds
    - tightness of proof - while rebuttal doesn't provide lowerbound it shows that their analysis is tight
    - practicality of VC theory on neural nets - rebuttal provides partial work-arounds
    - different $\epsilon_i$ - addresed
    - more realistic FL experiments - not addresed

**Reviewer Scores:**

- A reasonable Reviewer qTMm would have increased from current 4 to **6**: the rebuttal partially addresses Reviewer qTMm's concerns, but some fundamental issues regarding real world applicability hold.
- Reviewer gnyW would have likely retained their **4** since most of the concerns raised were fundamental.
- Reviewer hAX9 would have likely retained their **8**.
- Reviewer 4jxP would have likely retained **6**.

---

### Decision · Program_Chairs · 2026-01-26

Accept (Poster)